# An N-terminal motif in NLR immune receptors is functionally conserved across distantly related plant species

Hiroaki Adachi[1], Mauricio P Contreras[1], Adeline Harant[1], Chih-hang Wu[1], Lida Derevnina[1], Toshiyuki Sakai[1], Cian Duggan[2], Eleonora Moratto[2], Tolga O Bozkurt[2], Abbas Maqbool[1], Joe Win[1], Sophien Kamoun[1]*

[1]The Sainsbury Laboratory, University of East Anglia, Norwich Research Park, Norwich, United Kingdom; [2]Department of Life Sciences, Imperial College London, London, United Kingdom

**Abstract** The molecular codes underpinning the functions of plant NLR immune receptors are poorly understood. We used in vitro Mu transposition to generate a random truncation library and identify the minimal functional region of NLRs. We applied this method to NRC4—a helper NLR that functions with multiple sensor NLRs within a Solanaceae receptor network. This revealed that the NRC4 N-terminal 29 amino acids are sufficient to induce hypersensitive cell death. This region is defined by the consensus MADAxVSFxVxKLxxLLxxEx (MADA motif) that is conserved at the N-termini of NRC family proteins and ~20% of coiled-coil (CC)-type plant NLRs. The MADA motif matches the N-terminal α1 helix of Arabidopsis NLR protein ZAR1, which undergoes a conformational switch during resistosome activation. Immunoassays revealed that the MADA motif is functionally conserved across NLRs from distantly related plant species. NRC-dependent sensor NLRs lack MADA sequences indicating that this motif has degenerated in sensor NLRs over evolutionary time.

*For correspondence:
sophien.kamoun@tsl.ac.uk

## Introduction

Plants have evolved intracellular immune receptors to detect host-translocated pathogen virulence proteins, known as effectors (*Dodds and Rathjen, 2010*; *Jones et al., 2016*; *Kourelis and van der Hoorn, 2018*). These receptors, encoded by disease resistance (*R*) genes, are primarily nucleotide-binding, leucine-rich repeat proteins (NLRs). NLR-triggered immunity (also known as effector-triggered immunity) includes the hypersensitive response (HR), a type of programmed cell death associated with disease resistance. NLRs are widespread across eukaryotes and have been described in animals and fungi in addition to plants (*Jones et al., 2016*). In contrast to other taxa, plants express very large and diverse repertoires of NLRs, with anywhere from about 50 to 1000 genes encoded per genome (*Shao et al., 2016*; *Steuernagel et al., 2018*). Genome-wide analyses have defined repertoires of NLRs (NLRome) across plant species (*Shao et al., 2016*). An emerging paradigm is that plant NLRs form receptor networks with varying degrees of complexity (*Wu et al., 2018*). NLRs have probably evolved from multifunctional singleton receptors—which combine pathogen detection (sensor activity) and immune signalling (helper or executor activity) into a single protein—to functionally specialized interconnected receptor pairs and networks (*Adachi et al., 2019a*). However, our knowledge of the functional connections and biochemical mechanisms underpinning plant NLR networks remains limited. In addition, although dozens of NLR proteins have been subject to functional studies since their discovery in the 1990 s, this body of knowledge has not been interpreted through an evolutionary biology framework that combines molecular mechanisms with phylogenetics.

**eLife digest** Just like humans, plants get sick. They can be infected by parasites as diverse as fungi, bacteria, viruses, nematode worms and insects. But, also like humans, plants have an immune system that helps them defend against disease. Their first line of defence are disease resistance genes. Many of these genes encode so-called immune receptors, which are proteins that detect parasites and kick-off the immune response.

Plant genomes may encode anywhere between 50 and 1000 immune receptors; some of which work solo as singletons, while others operate in pairs or as complex networks. Understanding how immune receptor genes have evolved would give fundamental knowledge about how they work, which in turn would set the stage for researchers to be able to use them to protect agricultural crops from disease.

One driving force behind the evolution of many genes is gene duplication. Genes duplicate and afterwards the two copies can evolve in different ways. The original immune receptors are multi-tasking proteins that both detect parasites and trigger the immune response. Yet, following gene duplication, evolution has led to some immune receptors becoming dedicated to detection and losing the ability to trigger a defence response on their own.

Now, Adachi et al. have discovered a molecular signature – named the MADA motif – that defines the subset of immune receptors that can trigger the immune response in plants. This motif is made of just 21 amino acids (the building blocks of proteins) at one end of the receptor and, remarkably, a short fragment of the protein containing this motif is enough to trigger a defence response when produced in plants. In contrast, the immune receptors that have specialized to only detect parasites have lost this molecular signature throughout evolution, presumably because they do not need it as they rely on their receptor partners to trigger defences instead.

Every year, billions of dollars' worth of food is lost to plant diseases. These new findings will enable the research community to classify disease resistance genes into categories to help deduce the network architecture of the plant immune system. A better understanding of this, and how networks of plant immune receptor evolve, should set the stage for breeding crop plants that are more able to resist diseases.

NLRs are multidomain proteins of the ancient group of Signal Transduction ATPases (STAND) proteins that share a nucleotide-binding (NB) domain. In addition to the NB and LRR domains, most plant NLRs have characteristic N-terminal domains that define three subgroups: coiled-coil (CC), $CC_R$ or RPW8-like (RPW8) and toll and interleukin-1 receptor (TIR) (*Shao et al., 2016*). In metazoans, NLRs confer immunity to diverse pathogens through a wheel-like oligomerization process resulting in multiprotein platforms that recruit downstream elements, such as caspases (*Qi et al., 2010*; *Zhou et al., 2015*; *Hu et al., 2015*; *Zhang et al., 2015*; *Tenthorey et al., 2017*). Plant NLRs have long been thought to oligomerize through their N-terminal domains when they're activated (*Bentham et al., 2018*). However, the precise molecular mechanisms that underpin NLR activation and subsequent execution of HR cell death have remained largely unknown until very recently. In two remarkable papers, *Wang et al. (2019a)* and *Wang et al. (2019b)* have significantly advanced our understanding of both the structural and biochemical basis of CC-NLR activation in plants. They reconstituted the inactive and active complexes of the Arabidopsis CC-NLR ZAR1 (HOPZ-ACTI-VATED RESISTANCE1) with its partner receptor-like cytoplasmic kinases (RLCKs) (*Wang et al., 2019a*; *Wang et al., 2019b*). Cryo-electron microscopy (cryo-EM) structures revealed that activated ZAR1 forms a resistosome—a wheel-like pentamer that undergoes a conformational switch to expose a funnel-shaped structure formed by the N-terminal α helices (α1) of the CC domains (*Wang et al., 2019a*; *Wang et al., 2019b*). They propose an engaging model in which the exposed α1 helices of the ZAR1 resistosome mediate cell death by translocating into the plasma membrane and perturbing membrane integrity similar to pore-forming toxins (*Wang et al., 2019b*). However, whether the ZAR1 model extends to other CC-NLRs is unknown. One important unanswered question is the extent to which the α1 helix 'death switch' occurs in other CC-NLRs (*Adachi et al., 2019b*).

Although ZAR1 is classified as a singleton NLR that detects pathogen effectors without associating with other NLRs, many plant NLRs are interconnected in NLR pairs or networks (*Wu et al., 2018*; *Adachi et al., 2019a*). Paired and networked NLRs consist of sensor NLRs that detect pathogen effectors and helper NLRs that translate this effector recognition into HR cell death and immunity. In the Solanaceae, a major phylogenetic clade of CC-NLRs forms a complex immunoreceptor network in which multiple helper NLRs, known as NLR-REQUIRED FOR CELL DEATH (NRC), are required by a large number of sensor NLRs, encoded by *R* gene loci, to confer resistance against diverse pathogens, such as viruses, bacteria, oomycetes, nematodes and insects (*Wu et al., 2017*). These proteins form the NRC superclade, a well-supported phylogenetic cluster divided into the NRC helper clade (NRC-helpers or NRC-H) and a larger clade that includes all known NRC-dependent sensor NLRs (NRC-sensors or NRC-S) (*Wu et al., 2017*). The NRC superclade has expanded over 100 million years ago (Mya) from an NLR pair that diversified to up to one-half of the NLRs of asterid plants (*Wu et al., 2017*). How this diversification has impacted the biochemical activities of the NRC-S compared to their NRC-H mates is poorly understood. For example, it's unclear how the ZAR1 conceptual framework applies to more complex NLR configurations such as the NRC network (*Adachi et al., 2019b*).

This paper originates from use of the in vitro Mu transposition system to generate a random truncation library and identify the minimal region required for CC-NLR-mediated cell death. We applied this method to NRC4—a CC-NLR helper of the NRC family that is genetically required by a multitude of NRC-S, such as the potato late blight resistance protein Rpi-blb2, to cause HR cell death and confer disease resistance (*Wu et al., 2017*). This screen revealed that the N-terminal 29 amino acids of NRC4 are sufficient to induce cell death. Remarkably, this region is about 50% identical to the N-terminal ZAR1 $\alpha$1 helix, which undergoes the conformational 'death switch' associated with the activation of the ZAR1 resistosome (*Wang et al., 2019b*). Computational analyses revealed that this region is defined by a motif, following the consensus MADAxVSFxVxKLxxLLxxEx, which we coined the 'MADA motif'. This sequence is conserved not only in NRC4 and ZAR1 but also in ~20% of all CC-NLRs of dicot and monocot species. Motif swapping experiments revealed that the MADA motif is functionally conserved between NRC4 and ZAR1, as well as between NLRs from distantly related plant species. Interestingly, NRC-S lack N-terminal MADA sequences, which may have become nonfunctional over evolutionary time. We conclude that the evolutionarily constrained MADA motif is critical for the cell death inducing activity of CC domains from a significant fraction of plant NLR proteins, and that the 'death switch' mechanism defined for the ZAR1 resistosome is probably widely conserved across singleton and helper CC-NLRs.

## Results

### Mu mutagenesis of NRC4 reveals a short 29 amino acid N-terminal region that is sufficient for induction of HR cell death

The N-terminal CC domain of a subset of CC-NLR proteins can mediate self-association and trigger HR cell death when expressed on its own (*Bentham et al., 2018*). However, to date truncation experiments have been conducted based on educated guesses of domain boundaries (*Maekawa et al., 2011*; *Casey et al., 2016*; *Cesari et al., 2016*; *Wróblewski et al., 2018*). Moreover, one amino acid difference in the length of the assayed truncation can affect cell death inducing activity (*Casey et al., 2016*). Therefore, we designed an unbiased truncation approach using bacteriophage Mu in vitro transposition system to randomly generate a C-terminal deletion library of the helper NLR NRC4. By using a custom-designed artificial transposon (Mu-STOP transposon) that carries staggered translation stop signals at Mu R-end (*Poussu, 2005*), we targeted the full-length coding sequence of the NRC4 autoactive mutant, NRC4$^{D478V}$, (referred to from here on as NRC4$^{DV}$). We generated a total of 65 truncated NRC4$^{DV}$::Mu-STOP variants and expressed these mutants in *Nicotiana benthamiana* leaves using agroinfiltration (*Figure 1A*). Remarkably, only a single truncate carrying the N-terminal 29 amino acids triggered visible cell death in *N. benthamiana* leaves (*Figure 1B*, *Figure 1—figure supplement 1*). To validate this phenotype, we expressed NRC4 N-terminal 29 amino acids (NRC4$_{1-29}$) fused with the yellow fluorescent protein (YFP) at the C-terminus in *N. benthamiana* leaves (*Figure 2A*). NRC4$_{1-29}$-YFP triggered a visible cell death response, although the cell death intensity was weaker than that of the full-length NRC4$^{DV}$-YFP (*Figure 2B–E*).

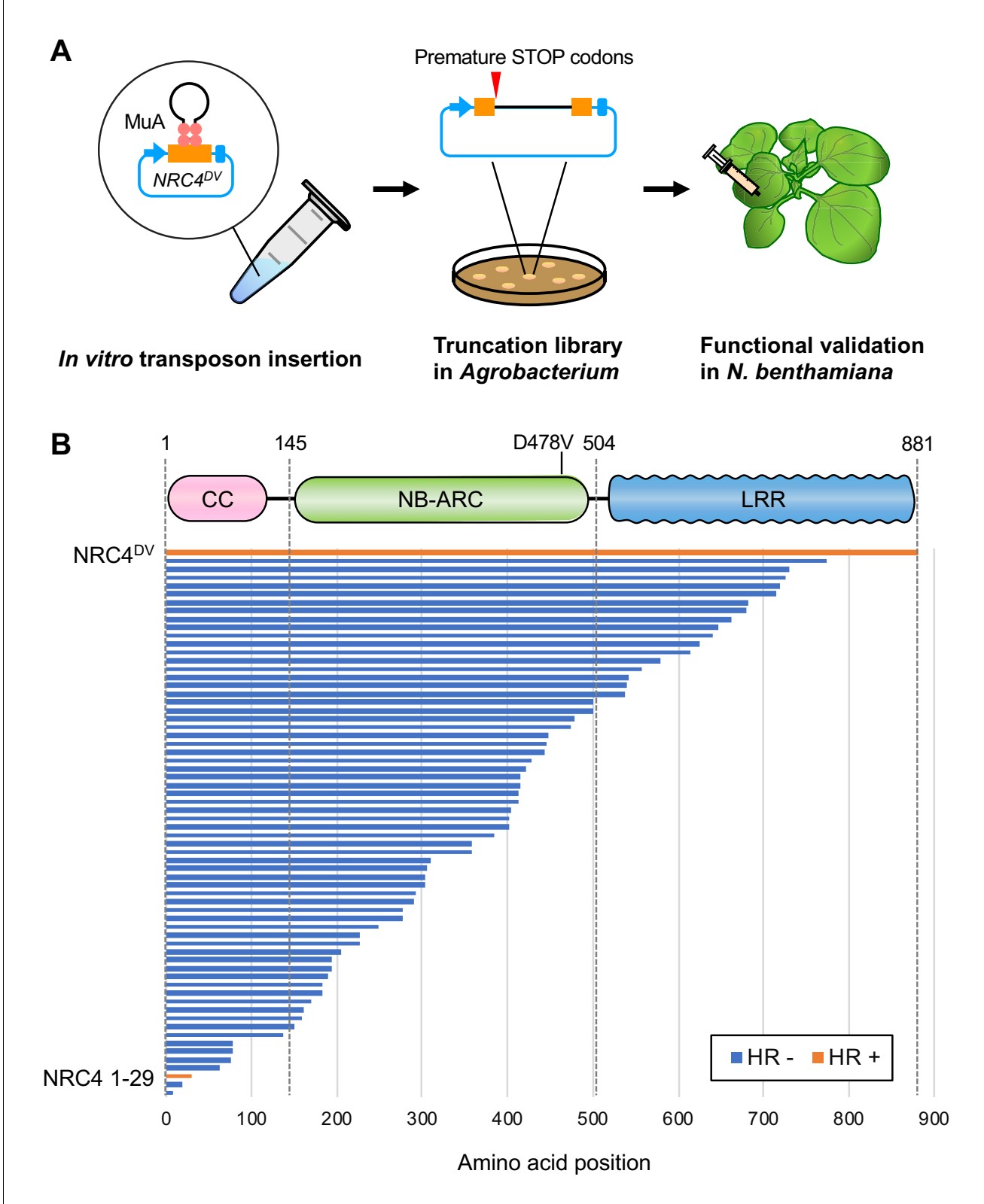

**Figure 1.** Transposon-based truncation mutagenesis reveals a short 29 amino-acid region sufficient for NRC4-mediated cell death. (**A**) Overview of the strategy for transposon-based C-terminal random truncation of NRC4 proteins. Hairpin Mu-STOP transposon and MuA proteins forming Mu transpososome were used for in vitro transposition into target plasmid. The truncation libraries (NRC4$^{DV}$::Mu-STOP) were transformed into *Agrobacterium* for transient expression in *N. benthamiana* leaves. The tube, petri dish and syringe are not drawn to scale. (**B**) NRC4$_{1-29}$::Mu-STOP

*Figure 1 continued on next page*

*Figure 1 continued*

triggers cell death in *N. benthamiana* leaves. In total, 65 truncated variants of NRC4$^{DV}$ were expressed in *N. benthamiana* leaves, and the cell death activities are described as cell death induction (orange, HR+) and no visible response (blue, HR-).

The online version of this article includes the following source data and figure supplement(s) for figure 1:

**Source data 1.** Sequences of NRC4 truncation library.

**Figure supplement 1.** Images of *N. benthamiana* leaves expressing truncated NRC4$^{DV}$::Mu-STOP variants.

To determine whether NRC4$_{1-29}$-YFP requires the endogenous *N. benthamiana* NRC4 to trigger cell death, we expressed this fusion protein in two independent mutant *nrc4a/b* plants that carry CRISPR/Cas9-induced mutations in the two NRC4 genes *NRC4a* and *NRC4b* (*Figure 2—figure supplement 1*, see Materials and methods). In these plants, NRC4$_{1-29}$-YFP still induced cell death indicating that the activity of the N-terminal 29 amino acids of NRC4 is independent of a full-length NRC4 protein (*Figure 2C–D and F–G*).

The CC domains of ZAR1 and maize Rp1 (RESISTANCE to PUCCINIA 1) are autoactive when expressed as a fusion protein with a fluorescent protein tag (*Wang et al., 2015*; *Baudin et al., 2017*). Given that YFP and related fluorescent proteins self-oligomerize (*Kim et al., 2015*), we hypothesized that such fluorescent proteins promote self-assembly of the N-terminal 29 amino acids of NRC4 resulting in hypersensitive cell death. To test this hypothesis, we modified YFP with the alanine 206 (A206) to lysine (K) mutation that reduces homo-affinity (*Figure 2—figure supplement 2A*) (*Zacharias et al., 2002*). The YFP$^{A206K}$ mutation compromised the cell death intensity of NRC4$_{1-29}$-YFP but not that of full-length NRC4$^{DV}$ (*Figure 2—figure supplement 2B–E*). This result indicates that YFP-mediated self-assembly is a key step in the capacity of NRC4$_{1-29}$-YFP to trigger hypersensitive cell death.

## NRC4 carries N-terminal sequences that are conserved across distantly related CC-NLRs

Our finding that the N-terminal 29 amino acids of NRC4 are sufficient to trigger cell death prompted us to investigate the occurrence of this sequence across the plant NLRome. We first compiled a sequence database containing 988 putative CC-NLRs and CC$_R$-NLRs (referred to from here on as CC-NLR database, *Figure 3A*, *Figure 3—figure supplement 1*) from six representative plant species (Arabidopsis, sugar beet, tomato, *N. benthamiana*, rice and barley) amended with 23 functionally characterized NLRs. Next, we extracted their sequences prior to the NB-ARC domain (*Figure 3A*). These sequences were too diverse and aligned poorly to each other to enable global phylogenetic analyses. Therefore, to classify the extracted N-terminal sequences based on sequence similarity, we clustered them into protein families using Markov cluster (MCL) algorithm Tribe-MCL (*Enright et al., 2002*) (*Figure 3A*). The 988 proteins clustered into 59 families of at least two sequences (tribes) and 43 singletons (*Figure 3B*). The largest tribe, Tribe 1, consists of 219 monocot NLRs, including MLA10, Sr33, Sr50, the paired Pik and Pia (RGA4 and RGA5) NLRs, and seven dicot NLRs notably RPM1 (*Figure 3B*). Tribe 2, the second largest tribe with 102 proteins, consists primarily of dicot proteins (93 out of 102) but still includes nine monocot NLRs. Interestingly, Tribe 2 grouped NRC-H proteins, including NRC4, with well-known CC-NLRs, such as ZAR1, RPP13, R2 and Rpi-vnt1.3 indicating that these proteins share similarities in their CC domains (*Figure 3B*).

We performed phylogenetic analyses of NLR proteins using the NB-ARC domain because it is the only conserved domain that produces reasonably good global alignments and can inform evolutionary relationships between all members of this family (*Figure 3—figure supplement 2*). We mapped individual NLR proteins grouped in Tribe-MCL N-terminal tribes onto a phylogenetic tree based on the NB-ARC domain (*Figure 3C*). These analyses revealed that the clustering of NLRs into the N-terminal tribes does not always match the NB-ARC phylogenetic clades (*Figure 3C*). In particular, NLRs in Tribe 1 and Tribe 2 often mapped to distinct well-supported clades scattered throughout the NB-ARC phylogenetic tree. We conclude that there are N-terminal domain sequences that have remained conserved over evolutionary time across distantly related CC-NLRs.

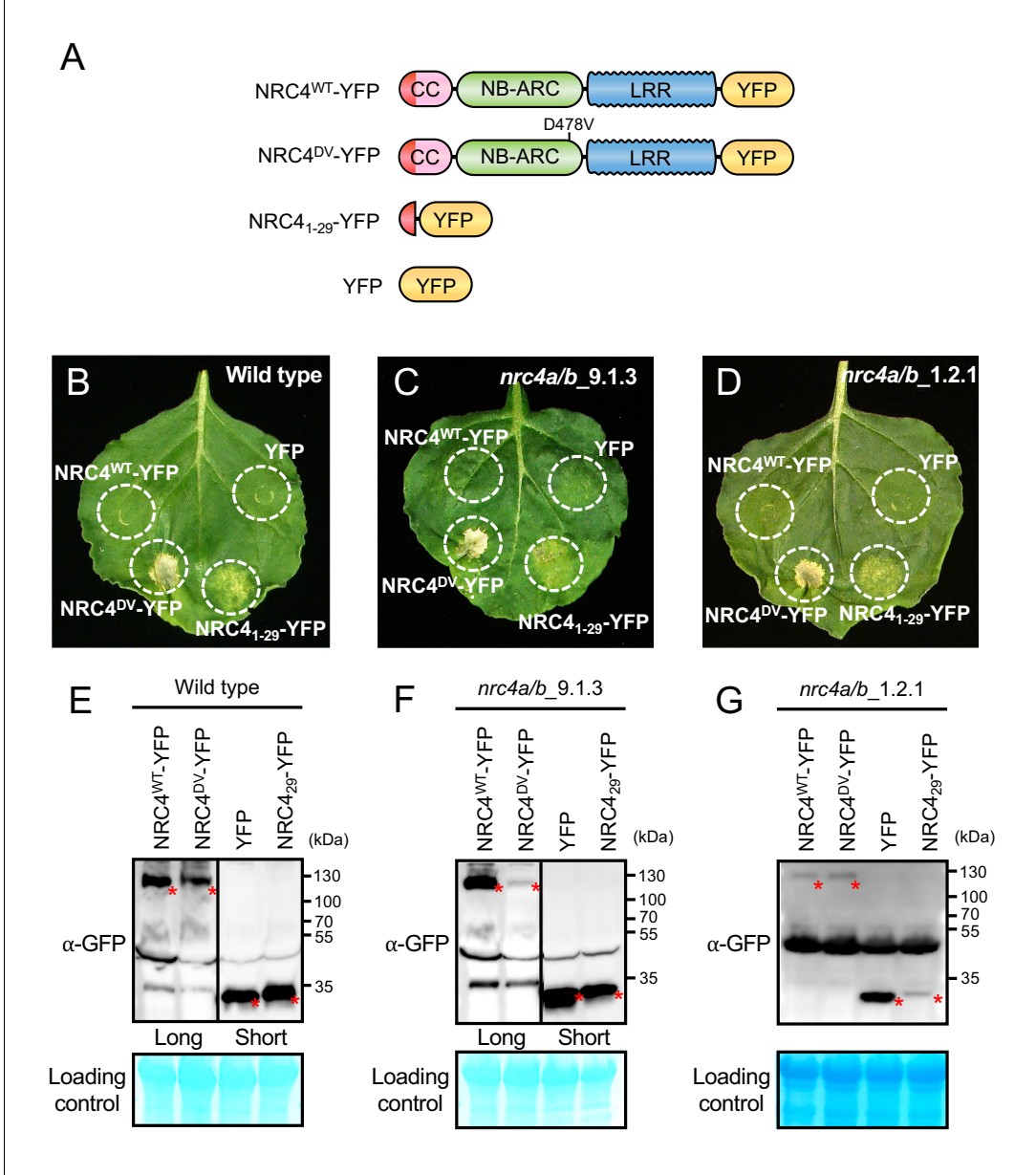

**Figure 2.** NRC4$_{1-29}$-YFP induces cell death in *Nicotiana benthamiana* independently of endogenous NRC4. (A) Schematic representation of wild-type NRC4-YFP (NRC4$^{WT}$-YFP) and the variants used for the in planta expression assays. The colour code is: red represents NRC4 1 to 29 amino acid region. (B) NRC4$_{1-29}$-YFP triggers cell death in wild-type *N. benthamiana* leaves. NRC4$^{WT}$-YFP, NRC4$^{DV}$-YFP, NRC4$_{1-29}$-YFP and YFP were co-expressed with the gene silencing suppressor p19 and photographed at 7 days after agroinfiltration. (C, D) NRC4$_{1-29}$-YFP triggers cell death in *N. benthamiana* independently of endogenous NRC4. Leaves of two independent *N. benthamiana nrc4a/b* lines were used for agroinfiltration assays as described in B. (E, F, G) Anti-GFP immunoblots of NRC4$^{WT}$-YFP, NRC4$^{DV}$-YFP, NRC4$_{1-29}$-YFP and YFP expressed in *N. benthamiana* wild-type and *nrc4a/b* mutants. Total proteins were prepared from wild-type and *nrc4a/b N. benthamiana* leaves at 1 day after agroinfiltration. Given that the full-length NLRs accumulate at much lower levels than the shorter peptide, we showed different exposures as indicated by the black line. Red asterisks indicate expected band sizes.

The online version of this article includes the following figure supplement(s) for figure 2:

**Figure supplement 1.** Knocking out of *NRC4a* and *NRC4b* in *Nicotiana benthamiana* impairs Rpi-blb2-mediated HR cell death.
**Figure supplement 2.** NRC4$_{1-29}$-YFP cell death is compromised by YFP A206K mutation.

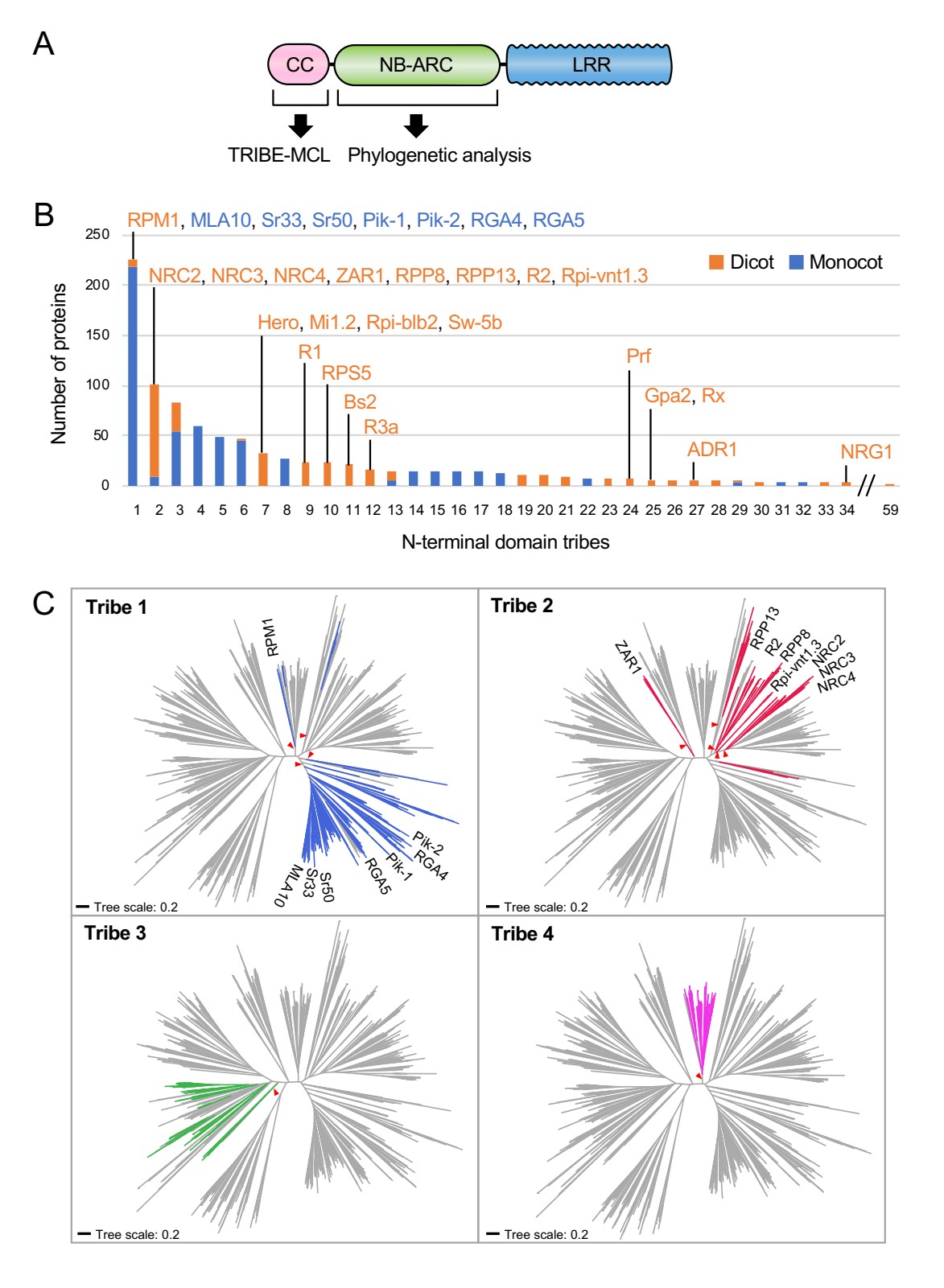

**Figure 3.** NRC4 carries N-terminal sequences that are conserved across distantly related CC-NLRs. (**A**) Schematic representation of the different NLR domains used in TRIBE-MCL and phylogenetic analyses. (**B**) Distribution of plant NLRs across N-terminal domain tribes. The colour codes are: orange for dicot NLRs and blue for monocot NLRs. (**C**) NLRs from the same N-terminal tribe are dispersed across NLR phylogeny. The phylogenetic tree was generated in MEGA7 by the neighbour-joining method using the NB-ARC domain sequences of 988 CC-NLRs identified from *N. benthamiana*, tomato, *Figure 3 continued on next page*

*Figure 3 continued*

sugar beet, Arabidopsis, rice and barley. Tribe 1 to Tribe 4 members are marked with different colours as indicated in each panel. Red arrow heads indicate bootstrap support >0.7 and is shown for the relevant nodes. The scale bars indicate the evolutionary distance in amino acid substitution per site. The full phylogenetic tree can be found in *Figure 3—figure supplement 2*.

The online version of this article includes the following source data and figure supplement(s) for figure 3:

**Source data 1.** Amino acid sequences of full-length NLRs in the CC-NLR database.
**Source data 2.** Amino acid sequences of N-terminal domains in the CC-NLR database.
**Source data 3.** N-terminal domain tribes of CC-NLRs.
**Figure supplement 1.** Phylogenetic analysis of NLR proteins from dicot and monocot plant species.
**Figure supplement 1—source data 1.** Amino acid sequences for CC/TIR-NLR phylogenetic tree.
**Figure supplement 1—source data 2.** CC/TIR-NLR phylogenetic tree file.
**Figure supplement 2.** Phylogenetic analysis of CC-NLR proteins from dicot and monocot plant species.
**Figure supplement 2—source data 1.** Amino acid sequences for CC-NLR phylogenetic tree.
**Figure supplement 2—source data 2.** CC-NLR phylogenetic tree file.

## NRC4 and ZAR1 share the N-terminal MADA motif

Next, we investigated whether N-terminal domains of CC-NLRs carry specific sequence motifs. We used MEME (Multiple EM for Motif Elicitation) (*Bailey and Elkan, 1994*) to identify conserved patterns in each of the N-terminal domain tribes. MEME revealed several conserved sequence patterns in each of the four largest tribes (*Figure 4—figure supplement 1*). The previously reported sequence pattern, EDVID motif (*Rairdan et al., 2008*), was as expected predicted in ~87% to 96% in the four largest tribes (*Figure 4—figure supplement 1*). Within Tribe 2, a motif that is conserved at the N terminus of 87 of 102 proteins overlapped with the N-terminal 29 amino acids of NRC4 we identified as sufficient to cause cell death (*Figure 4—figure supplement 1*). Remarkably, the conserved sequence pattern of this very N-terminal motif matched the ZAR1 α1 helix that undergoes a conformational switch during activation of the ZAR1 resistosome (*Wang et al., 2019b*) (*Figure 4A–B*). In fact, 8 of the first 17 amino acids of ZAR1 are invariant in NRC4, and the majority of the amino acid polymorphisms between ZAR1 and NRC4 in the α1 helix region are conservative (*Figure 4A*). We conclude that NRC4, ZAR1 and numerous other CC-NLRs share a conserved N-terminal motif. We coined this sequence 'MADA motif' based on the deduced 21 amino acid consensus sequence MADAxVSFxVxKLxxLLxxEx (*Figure 4A*, *Figure 4—figure supplement 2*).

## The MADA motif is primarily found in NLR proteins

We built a Hidden Markov Model (HMM) from a sequence alignment of the MADA motif of 87 NLR proteins from Tribe 2. To determine whether the MADA motif is primarily found among proteins annotated as NLRs, we used the HMMER software (*Eddy, 1998*) to query the Arabidopsis and tomato proteomes using the MADA motif HMM. HMMER searches revealed that the MADA motif is mainly found in NLR proteins compared with non-NLR proteins (*Figure 4C*). An HMM score cut-off of 10.0 clearly distinguishes NLR proteins from others with 97.1% (34 out of 35) tomato proteins and 97.7% (42 out of 43) Arabidopsis proteins scoring over 10.0 being annotated as NLRs (*Figure 4C*). We conclude that the MADA motif is a sequence signature of a subset of NLR proteins and that a HMMER cut-off score of 10.0 is most optimal for high confidence searches of MADA containing CC-NLR proteins (MADA-CC-NLRs).

## MADA-like sequences occur in the N-termini of about 20% of dicot and monocot CC-NLRs

To what extent does the MADA motif occur in plant NLRomes? We re-screened the CC-NLR database using HMMER and identified 103 hits (10.4%) over the cut-off score of 10.0 (*Figure 5A–B*, *Figure 5—figure supplement 1A*). We also noted that another 129 NLRs were positive but with a score lower than 10.0, and we tentatively termed these hits as MADA-like CC-NLRs (MADAL-CC-NLRs) (*Figure 5B*, *Figure 5—figure supplement 1A*). Most of the MADA hits are from dicot plant species whereas MADAL-CC-NLRs are primarily from monocots possibly reflecting a bias in our HMM profile which was built from the dicot enriched Tribe 2 (*Figure 5C*, *Figure 5—figure supplement 1B*). Indeed, the majority of MADA hits (85 out of 103) were from Tribe 2, which includes NRC4 and

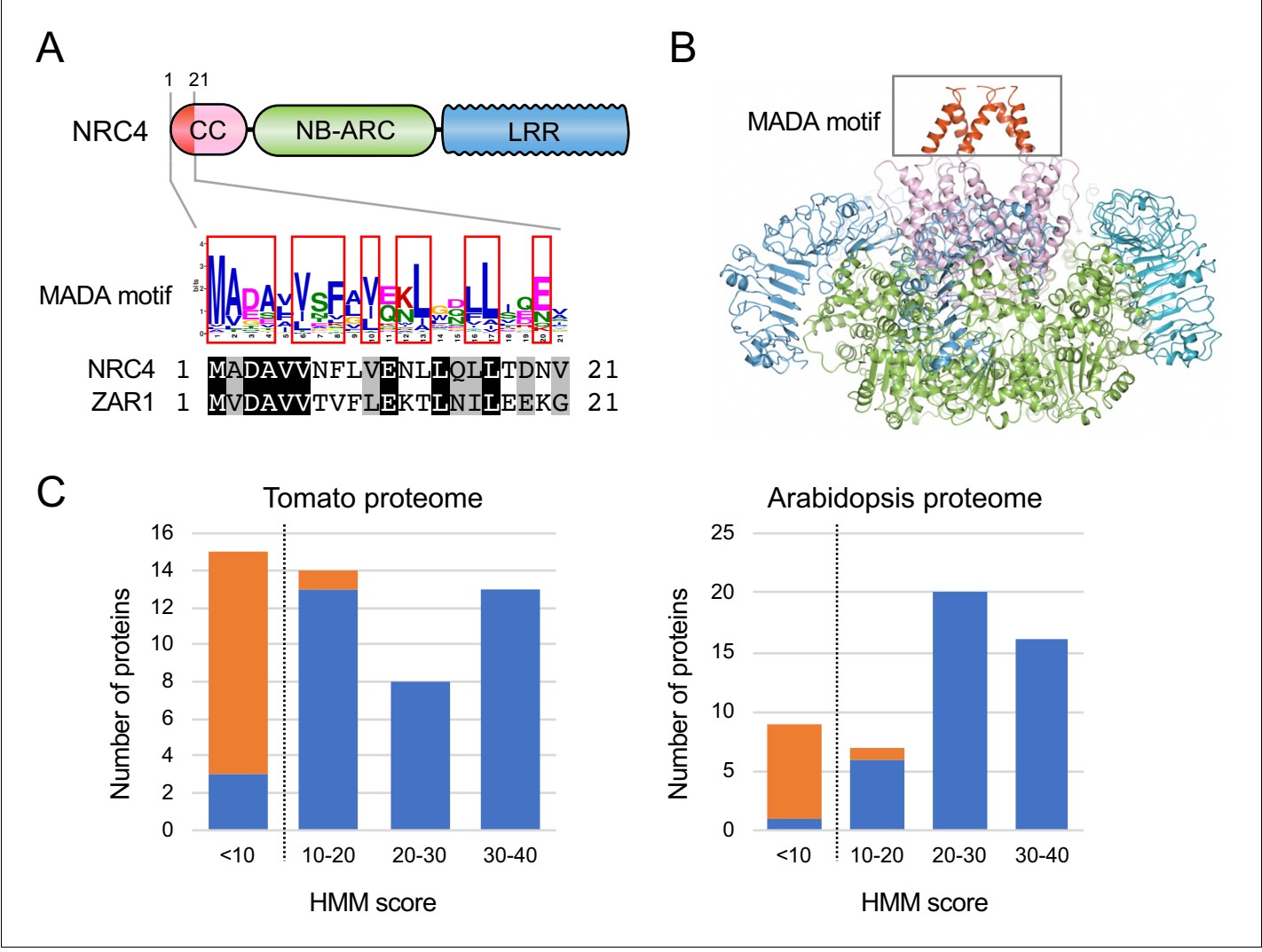

**Figure 4.** The MADA motif is a conserved unit at the very N-terminus of NRC4 and ZAR1. (**A**) Schematic representation of a classical CC-NLR protein highlighting the position of the MADA motif. Consensus sequence pattern of the MADA motif identified by MEME along with an alignment of NRC4 and ZAR1. Red boxes refer to residues conserved over 45% in Tribe 2 NLRs. (**B**) A structure homology model of NRC4 based on ZAR1 resistosome illustrating the position of the MADA motif. Each of the modelled five monomers is illustrated in cartoon representation. The colour code is: red for the MADA motif. The grey box highlights the N-terminal α helices, which contain the MADA motif. (**C**) Distribution of the MADA motif in tomato (left) and Arabidopsis (right) proteomes following HMMER searches with the MADA motif HMM. The number of proteins in each HMM score bin is shown. NLR and non-NLR proteins are shown in blue and orange, respectively. The dashed line indicates the cut-off used to define the most robust MADA-CC-NLR. NLRs with scores < 10.0 were classified as MADA-like NLRs (MADAL-NLRs).

The online version of this article includes the following source data and figure supplement(s) for figure 4:

**Source data 1.** Output of the HMMER search using the MADA motif HMM against tomato and Arabidopsis proteomes.
**Source data 2.** Amino acid sequences of the MADA motif.
**Figure supplement 1.** CC-NLRs have conserved protein sequence patterns in the beginning of the N-terminal domains.
**Figure supplement 2.** N terminus of NRC4 possesses a consensus pattern coined MADA motif.

ZAR1, but some MADA hits were also from other Tribes, notably the rice helper NLR Pik-2 from Tribe 1 (HMM score = 10.4) (*Figure 5C*, *Figure 5—figure supplement 1C*). MADAL-CC-NLRs are mainly from Tribe 1 and Tribe 4 and include the monocot proteins MLA10 and Sr33, as well as Arabidopsis RPM1 (*Figure 5C*, *Figure 5—figure supplement 1C*).

Given that the MADA sequence is at the very N-terminus of ZAR1 and NRC4, and that the N-terminal position of the ZAR1 α1 helix is critical for its function based on the model of *Wang et al. (2019b)*, we checked the positional distribution of predicted MADA and MADAL motifs (*Figure 5D*).

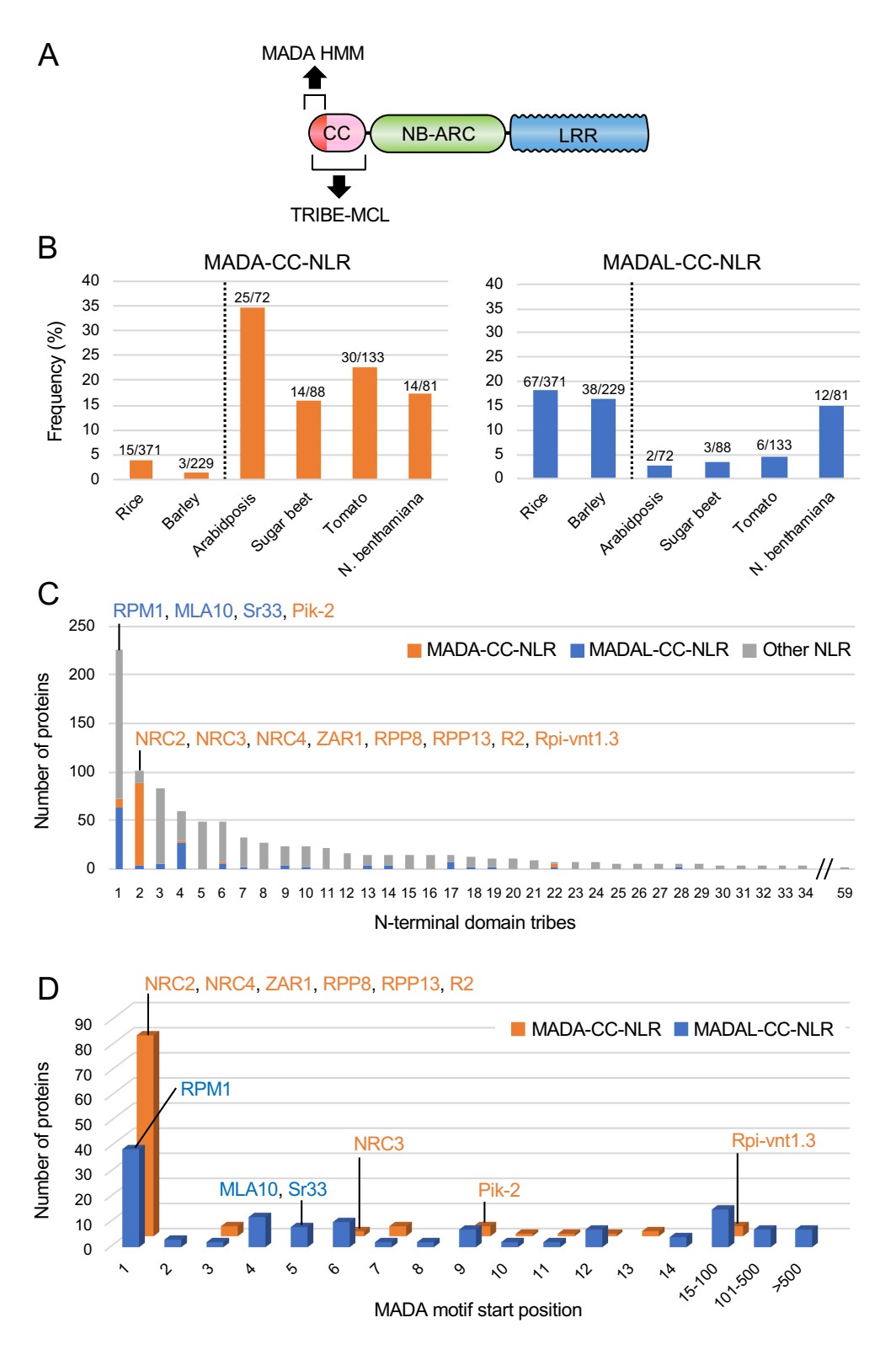

**Figure 5.** The MADA motif is conserved in ~20% of CC-NLRs. (**A**) Schematic representation of a classical CC-NLR protein highlighting the regions used for HMMER searches (MADA-HMM) and for TRIBE-MCL. (**B**) Occurrence of MADA/MADAL-CC-NLRs in representative species of monocots and dicots. The frequency of MADA/MADAL-CC-NLRs for each plant species was calculated as a percentage of all predicted CC-NLR proteins. (**C**) Occurrence of MADA/MADAL-CC-NLRs in N-terminal domain tribes of CC-NLRs. (**D**) Position distribution of MADA/MADAL motif relative to the start codon position

*Figure 5 continued on next page*

*Figure 5 continued*

among the identified 103 MADA-CC-NLRs and 129 MADAL-CC-NLRs. The colour codes are: orange for MADA-CC-NLRs, blue for MADAL-CC-NLRs and grey for other NLRs.

The online version of this article includes the following source data and figure supplement(s) for figure 5:

**Source data 1.** Output of the HMMER search using the MADA motif HMM against the CC-NLR database.
**Source data 2.** List of the predicted Arabidopsis MADA-CC-NLRs.
**Figure supplement 1.** Bar graph of MADA/MADAL-CC-NLRs according to HMM score.

The majority of the predicted MADA and MADAL motifs (199 out of 232, 85.8%) occurred at the very beginning of the NLR protein. However, 4 of 103 of the predicted MADA- and 29 of 129 MADAL-CC-NLRs have N-terminal extensions over 15 amino acids prior to the motifs (*Figure 5D*). For example, the MADA motif is located at position 54 to 72 amino acids in the potato NLR Rpi-vnt1.3. Whether these exceptions reflect misannotated gene models or genuinely distinct motif sequences remains to be determined.

In summary, our bioinformatic analyses revealed that 199 out 988 (20.1%) of the CC-NLRs of six representative dicot and monocot species contain a MADA or MADAL motif at their very N-termini. These MADA sequences have noticeable similarity to NRC4 and ZAR1.

## NRC-dependent sensor NLRs (NRC-S) lack the MADA motif

NB-ARC domain phylogenetic trees revealed that the NRC superclade is divided into the NRC clade (NRC-H) and a larger clade that includes all known NRC-dependent sensor NLRs (NRC-S) (*Wu et al., 2017*). We noted that even though the NRC-H and NRC-S are sister clades based on NB-ARC phylogenetic analyses, they grouped into distinct N-terminal domain tribes in the Tribe-MCL analyses (*Figure 6A*). Whereas all NRC-H mapped to Tribe 2, NRC-S clustered into eight different tribes (*Figure 6A*). This pattern indicates that unlike the NRCs, the N-terminal sequences of their NRC-S mates have diversified throughout evolutionary time.

Next, we mapped the occurrence of the MADA motif onto the NB-ARC phylogenetic tree and noted that the distribution of the MADA motif was uneven across the NRC superclade despite their phylogenetic relationship (*Figure 6B*). Whereas 20 out of 22 NRC-H have a predicted MADA motif at their N-termini, none of the 117 examined NRC-S were predicted as MADA-CC-NLR in the HMMER search (*Figure 6B*). In fact, 65 of 117 NRC-S, including the well know disease resistance proteins R1, Prf, Sw5b, Hero, Rpi-blb2 and Mi-1.2, have N-terminal extensions of ~600 amino acids, or more in the case of Prf, prior to their predicted CC domains (*Figure 6B*). These findings indicate the CC domains of NRCs and their NRC-S mates have experienced distinct evolutionary trajectories even though these NLR proteins share a common evolutionary origin.

## MADA motif residues are required for NRC4 to trigger cell death

To experimentally validate our bioinformatic analyses, we performed site directed mutagenesis to determine the degree to which the MADA motif is required for the activity of NRC4. First, we followed up on the ZAR1 structure-function analyses of *Wang et al. (2019b)* who showed that three amino acids (phenylalanine 9 [F9], leucine 10 [L10] and leucine 14 [L14]) within the α1 helix/MADA motif are required for ZAR1-mediated cell death and bacterial resistance. We introduced a triple alanine substitution similar to the mutant of *Wang et al. (2019b)* into the autoactive NRC4$^{DV}$ and found that this L9A/V10A/L14A mutation significantly reduced, but did not abolish, NRC4$^{DV}$ cell death inducing activity (*Figure 7A–C*). Given that the MADA motif, particularly the mutated L9, V10 and L14 sites, is primarily composed of hydrophobic residues, we reasoned that substitutions with the negatively charged glutamic acid (E) would be more disruptive than hydrophobic alanine. Therefore, we substituted L9, V10 and L14 with glutamic acid, and observed that the L9E/V10E/L14E mutation resulted in a more severe disruption of the cell death activity of NRC4$^{DV}$ compared to the triple alanine mutant (*Figure 7A–C*). Both of the NRC4$^{DV}$ triple alanine and glutamic acid mutant proteins accumulated to similar levels as NRC4$^{DV}$ when expressed in *N. benthamiana* leaves indicating that the observed loss-of-function phenotypes were not due to protein destabilization (*Figure 7D*).

We further introduced the triple alanine mutation to NRC4$_{1-29}$-YFP and ZAR1$_{1-144}$-YFP (*Figure 7—figure supplement 1A*). ZAR1$_{1-144}$ matches the ZAR1 CC domain and is known to trigger cell death

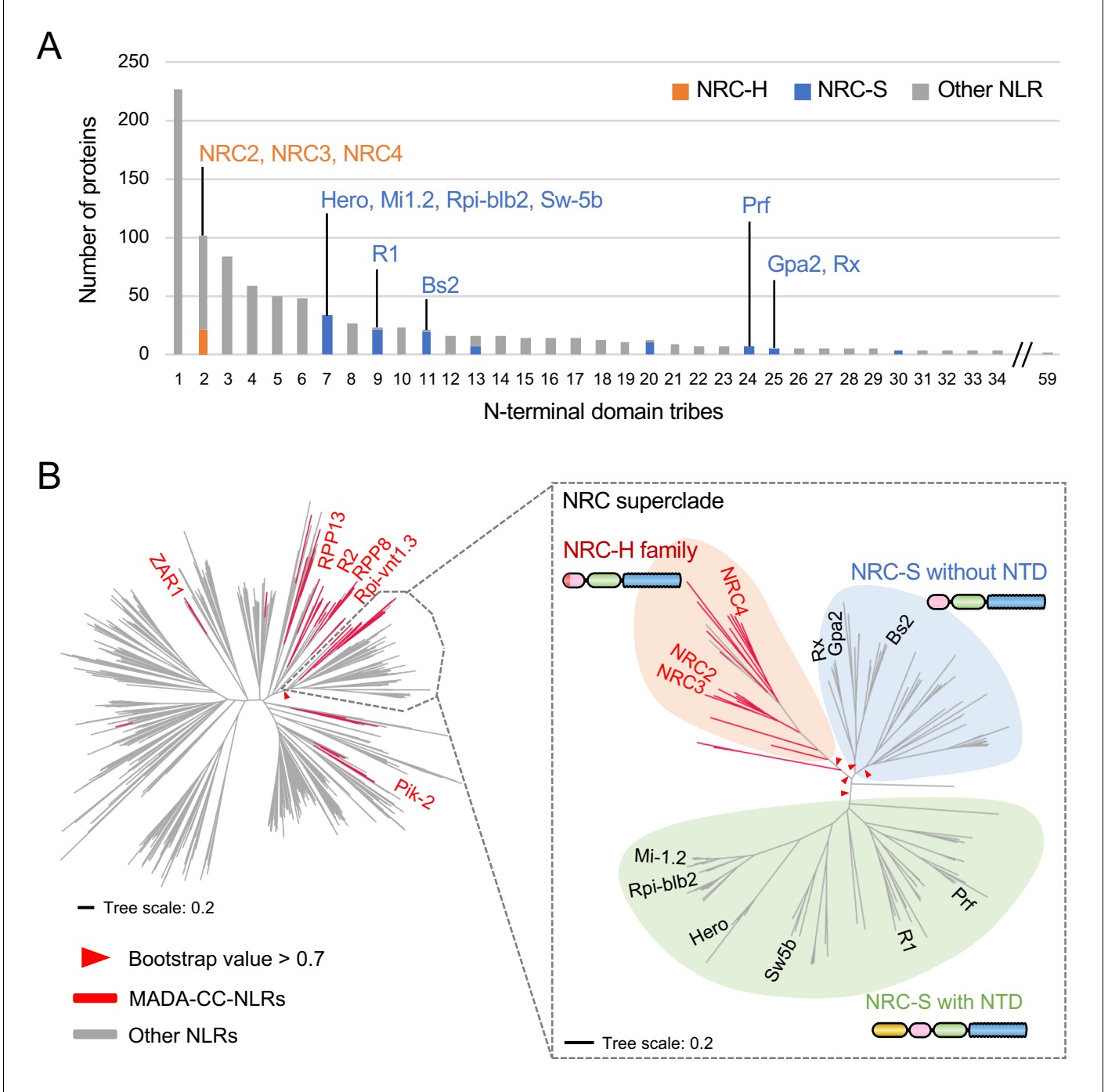

**Figure 6.** NRC-dependent sensors (NRC-S) do not have the MADA motif. (**A**) Distribution of NRCs (NRC-H) and NRC-dependent sensors (NRC-S) across N-terminal domain tribes of CC-NLRs. Individual NLR members of the NRC superclade were classified based on phylogenetic analysis as described in *Figure 3—figure supplement 2*. The colour codes are: orange for the NRCs (NRC-H), blue for the NRC-sensors (NRC-S) and grey for other NLRs. (**B**) NRC-dependent sensors (NRC-S) do not contain the MADA motif. The phylogenetic tree of the 988 CC-NLRs described in *Figure 3C* is shown in the left panel with the NRC superclade marked by the grey lines. The NRC superclade phylogenetic tree is shown on the right panel and highlights the well-supported subclades NRC-H and the expanded NRC-S. The NRC-S clade is divided into NLRs that lack an N-terminal extension domain (NTD) prior to their CC domain and those that carry an NTD. MADA-CC-NLRs are highlighted in red in both trees. Red arrowheads mark bootstrap supports >0.7 in relevant nodes. The scale bars indicate the evolutionary distance in amino acid substitution per site. The full phylogenetic tree can be found in *Figure 3—figure supplement 2*. Schematic representation of domain architecture of the depicted classes of NLR protein is also shown similar to the other figures but with the ~600 amino acid NTD shown in yellow.

The online version of this article includes the following source data for figure 6:

**Source data 1.** HMM scores of NRC-superclade proteins.

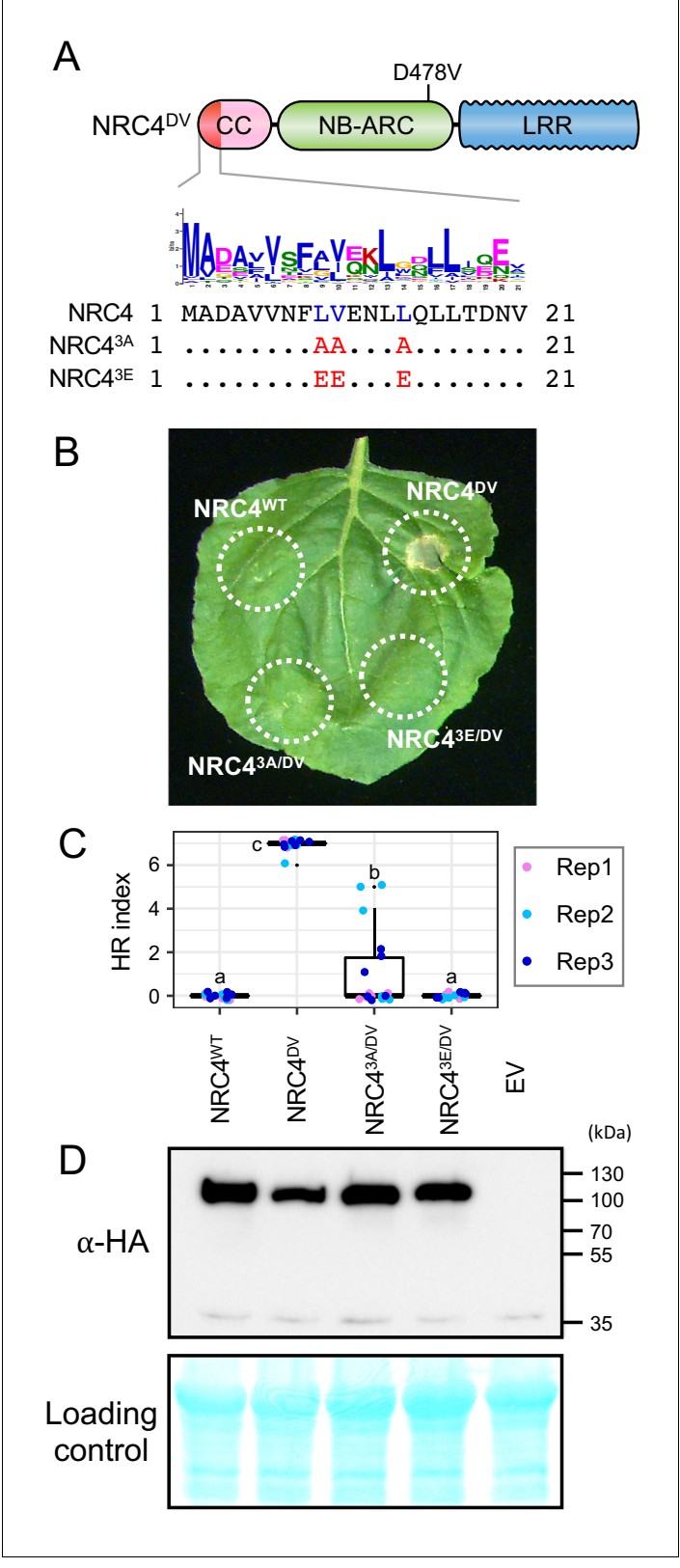

**Figure 7.** L9, V10 and L14 triple mutation impairs cell death activity of autoimmune NRC4$^{DV}$. (**A**) Schematic representation of NRC4 and the mutated sites in the MADA motif. Mutated sites and substituted residues are shown as red characters in the NRC4 sequence alignment. (**B**) Cell death observed in *N. benthamiana* after expression of NRC4 mutants. *N. benthamiana* leaf panels expressing NRC4$^{WT}$-6xHA, NRC4$^{DV}$-6xHA, NRC4$^{3A/DV}$-

*Figure 7 continued on next page*

*Figure 7 continued*

6xHA and NRC4$^{3E/DV}$-6xHA were photographed at 5 days after agroinfiltration. (**C**) Box plots showing cell death intensity scored as an HR index based on three independent experiments. Statistical differences among the samples were analysed with Tukey's honest significance difference (HSD) test (p<0.01). (**D**) In planta accumulation of the NRC4 variants. For anti-HA immunoblots of NRC4 and the mutant proteins, total proteins were prepared from *N. benthamiana* leaves at 1 day after agroinfiltration. Empty vector control is described as EV. Equal loading was checked with Reversible Protein Stain Kit (Thermo Fisher).

The online version of this article includes the following figure supplement(s) for figure 7:

**Figure supplement 1.** NRC4$_{1-29}$-YFP cell death is compromised by L9, V10 and L14 triple mutation.

when expressed fused to a YFP tag (*Baudin et al., 2017*). The triple alanine mutation abolished the cell death triggered by both NRC4$_{1-29}$-YFP and ZAR1$_{1-144}$-YFP, supporting the view that MADA motifs are essential for the capacity of the N-termini of NRC4 and ZAR1 to cause cell death (*Figure 7—figure supplement 1B–D*).

Next, we performed single alanine and glutamic acid mutant scans to reveal which other residues in the MADA motif are required for NRC4-mediated cell death. None of the tested single alanine-substituted mutants affected the cell death response of NRC4$^{DV}$ (*Figure 8—figure supplement 1*). In contrast, single glutamic acid mutations L9E, L13E and L17E essentially abolished the cell death activity of NRC4$^{DV}$ without affecting the stability of the mutant proteins (*Figure 8*). Therefore, we determined that the L9, L13, and L17 residues in the MADA motif are critical for cell death induction by NRC4.

Finally, we mapped L9, L13 and L17 onto a homology model of the CC domain of NRC4 produced based on the ZAR1 resistosome structure of *Wang et al. (2019b)* (*Figure 8—figure supplement 2*). All three residues mapped to the outer surface of the funnel-shaped structure formed by the α1 helices similar to the previously identified residues in positions 9, 10 and 14. These results suggest that the outer surface of the funnel-shaped structure formed by N-terminal helices is critical not only for the function of ZAR1 but also for the activity of another MADA-CC-NLR.

## NRC4$_{1-29}$-YFP forms MADA motif- and YFP-dependent puncta

The ZAR1 model postulates that the resistosome translocates into the plasma membrane through the α1 helix which matches the MADA motif (*Wang et al., 2019b*). To investigate the intracellular dynamics of the MADA motif, we analysed the subcellular distribution of NRC4$_{1-29}$-YFP in *N. benthamiana* leaves (*Figure 9A*). Interestingly, unlike free YFP which typically shows nucleocytoplasmic distribution, NRC4$_{1-29}$-YFP produced fluorescence signal in punctate structures throughout the cell in addition to relatively weak nucleocytoplasmic signal (*Figure 9A*). Furthermore, we merged both the z-stack and single plain images of the YFP proteins with the plasma membrane marker RFP-Rem1.3 (*Bozkurt et al., 2014*). Although the NRC4$_{1-29}$-YFP puncta did not completely overlap with RFP-Rem1.3 signal, we noticed some of the NRC4$_{1-29}$-YFP puncta associated with the plasma membrane (*Figure 9A–B*).

To further study the NRC4$_{1-29}$-YFP puncta, we examined puncta formation of the YFP A206K mutant, which shows reduced cell death by NRC4$_{1-29}$-YFP (*Figure 2—figure supplement 2*). In contrast to NRC4$_{1-29}$-YFP, NRC4$_{1-29}$-YFP$^{A206K}$ rarely formed puncta (*Figure 9A,C*), suggesting that YFP self-assembly is required for NRC4$_{1-29}$-YFP puncta formation. Furthermore, introducing the L9E in NRC4$_{1-29}$-YFP greatly reduced puncta formation (*Figure 9A,C*). This finding directly connects puncta formation to the activity of full length NRC4 given that L9E also affects NRC4 cell death activity (*Figure 8*). Taken together, these results indicate that both an intact MADA motif and YFP oligomerization are required for the capacity of NRC4$_{1-29}$-YFP to form puncta as well as cause cell death in *N. benthamiana* leaves.

## The α1 helix of arabidopsis ZAR1 and the N-termini of other MADA-CC-NLRs can functionally replace the N-terminus of NRC4

Our observation that the ZAR1 α1 helix has sequence similarity to the N-terminus of NRC4 prompted us to determine whether this sequence is functionally conserved between these two proteins. To test this hypothesis, we swapped the first 17 amino acids of NRC4$^{DV}$ with the equivalent

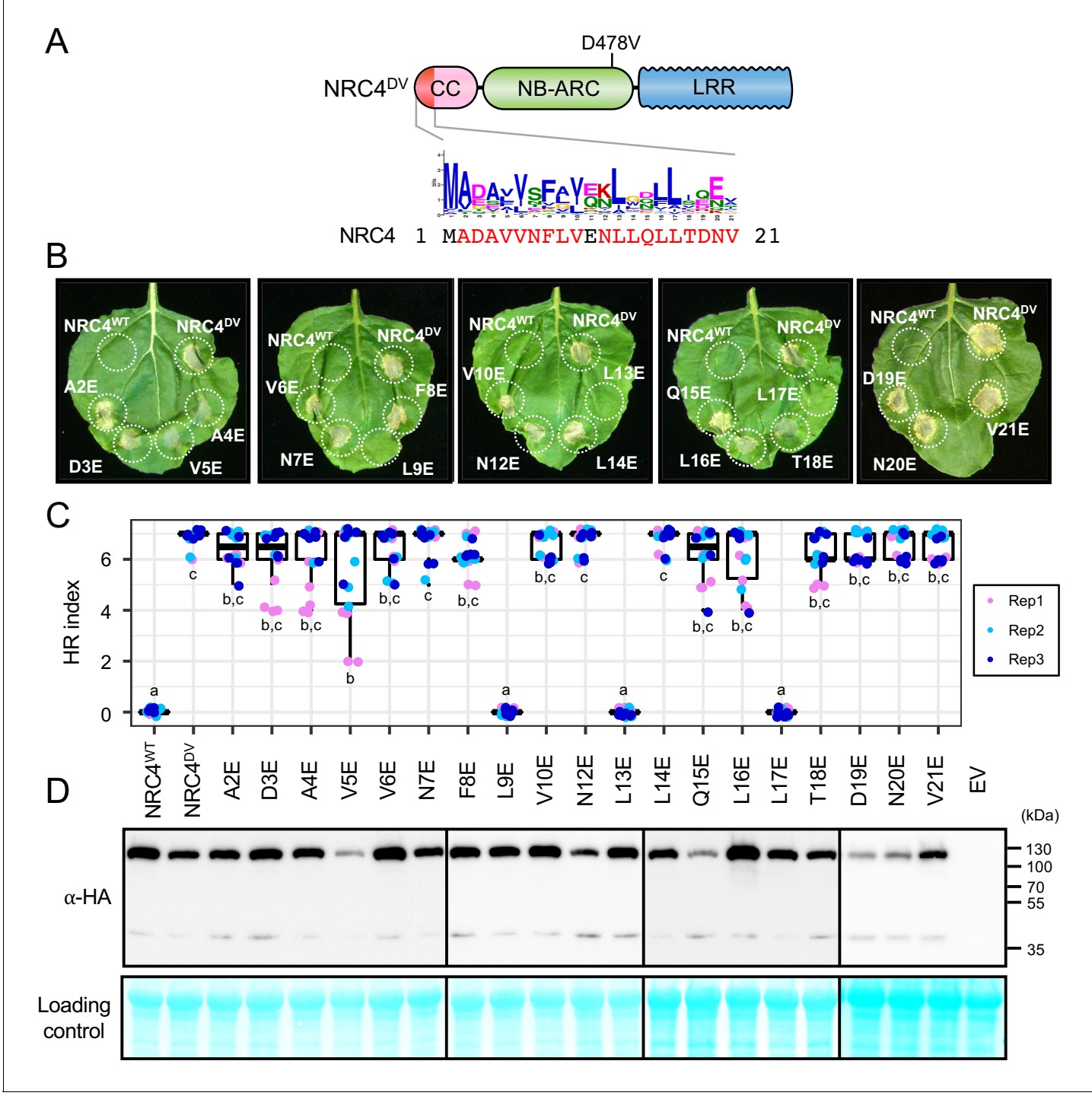

**Figure 8.** L9E, L13E and L17E single mutations impair cell death activity of autoimmune NRC4DV. (**A**) Schematic representation of NRC4 and the glutamic acid (**E**) mutant scan of the MADA motif. Mutated sites are shown as red characters in the NRC4 sequence. (**B**) Cell death observed in *N. benthamiana* after expression of NRC4 mutants. *N. benthamiana* leaf panels expressing NRC4WT-6xHA, NRC4DV-6xHA and the corresponding E mutants were photographed at 5 days after agroinfiltration. (**C**) Box plots showing cell death intensity scored as an HR index based on three independent experiments. Statistical differences among the samples were analysed with Tukey's HSD test (p<0.01). (**D**) In planta accumulation of the NRC4 variants. Immunoblot analysis was done as described in *Figure 7D*.

The online version of this article includes the following figure supplement(s) for figure 8:

**Figure supplement 1.** Alanine mutants do not compromise HR cell death triggered by autoactive NRC4.

**Figure supplement 2.** Mapping loss of function mutations on N-terminal α helices of NRC4.

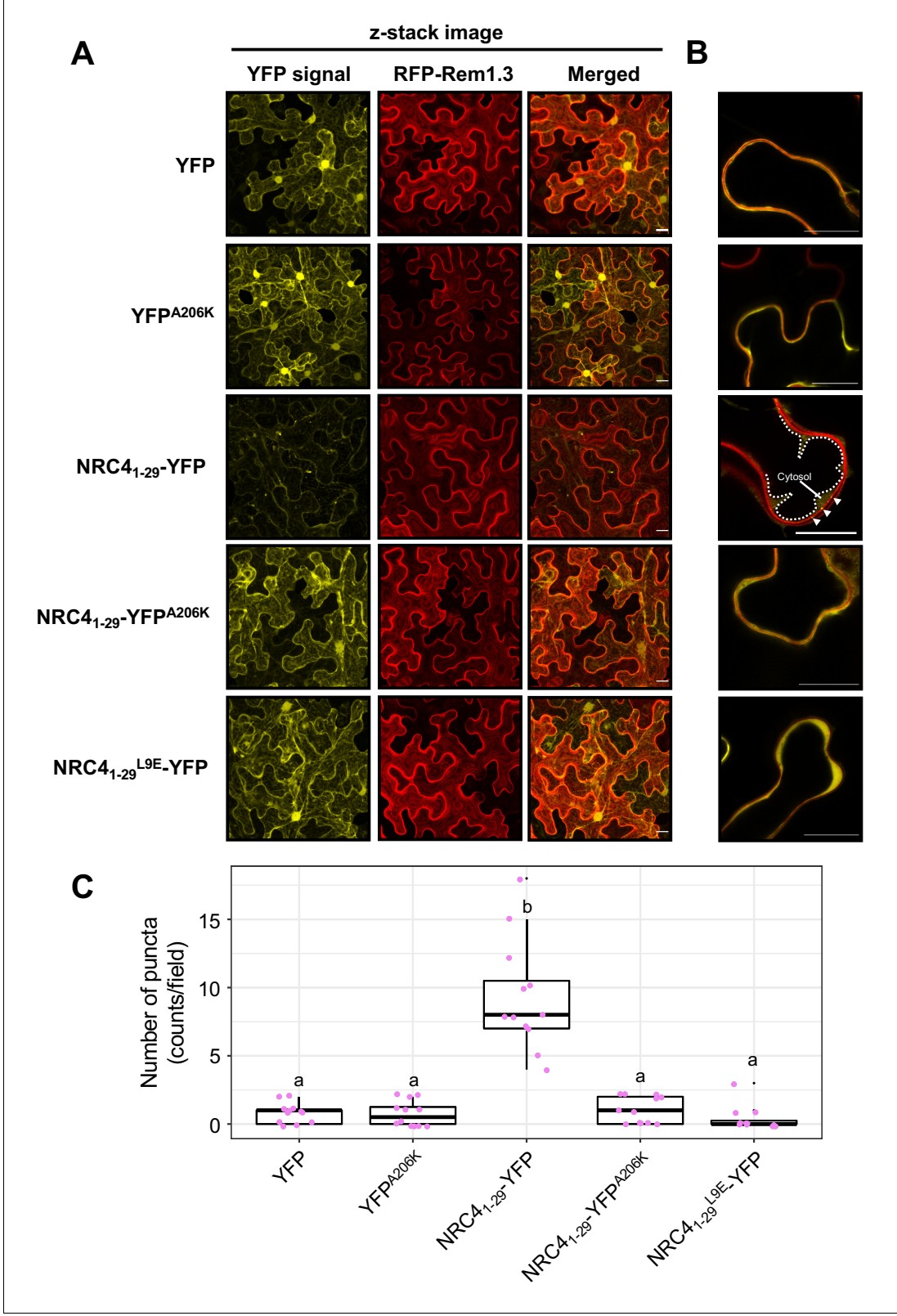

**Figure 9.** NRC4$_{1-29}$-YFP forms MADA- and YFP-dependent puncta. (**A**) Subcellular localization of NRC4$_{1-29}$-YFP and the mutant proteins in *N. benthamiana* epidermal cells. *N. benthamiana* leaves expressing YFP, YFP$^{A206K}$, NRC4$_{1-29}$-YFP, NRC4$_{1-29}$-YFP$^{A206K}$ and NRC4$_{1-29}^{L9E}$-YFP were imaged 2 days after agroinfiltration. (**B**) Single plain image of NRC4$_{1-29}$-YFP puncta. White dotted line indicates the tonoplast in *N. benthamiana* epidermal cell. White arrowheads point to NRC4$_{1-29}$-YFP puncta. Scale bars are 20 µm. (**C**) Quantification of puncta formation. The number of high intensity puncta was

*Figure 9 continued on next page*

*Figure 9 continued*

counted using maximum intensity Z-projection images from 12 independent observations. Statistical differences among the samples were analysed with Tukey's HSD test (p<0.01).

region of ZAR1 (*Figure 10A–B*). The resulting ZAR1$_{1-17}$-NRC4 chimeric protein can still trigger cell death in *N. benthamiana* leaves indicating that the MADA/α1 helix sequence is functionally equivalent between these two NLR proteins (*Figure 10C*, *Figure 10—figure supplement 1*).

Next, we swapped the same 17 amino acids of NRC4 with the matching sequences of the MADA-CC-NLRs NRC2 from *N. benthamiana*, RPP8 and RPP13 from Arabidopsis, and Pik-2 and Os03g30910.1 from rice, all of which gave HMMER scores > 10.0 and ranging from 30.8 to 10.4 (*Figure 10A–B*). All of the assayed chimeric NRC4$^{DV}$ proteins retained the capacity to trigger cell death in *N. benthamiana* leaves (*Figure 10C*, *Figure 10—figure supplement 1*). We determined whether the N-termini of MADAL-CC-NLRs Arabidopsis RPM1 and barley MLA10, which yielded respective HMMER scores of 9.3 and 7.8, could also replace the first 17 amino acids of NRC4$^{DV}$ (*Figure 10A–B*). Both NRC4$^{DV}$ chimeras retained the capacity to trigger cell death indicating that these MADAL sequences are functionally analogous to the NRC4 N-terminus (*Figure 10C*, *Figure 10—figure supplement 1*). These results indicate that the MADA motif is functionally conserved even between distantly related NLRs from dicots and monocots.

We further swapped the 17 amino acids of NRC4$^{DV}$ with N-terminal sequences from Arabidopsis LOV1 (AT1G10920), pepper Bs2 and potato Rx, all of which were not predicted to have a MADA sequences by HMMER searches (*Figure 10A–B*). LOV1 was among the 13.7% of Tribe 2 NLRs that were not predicted to have a MADA/MADAL motif. Bs2 and Rx are NRC-S NLRs that belong to different tribes—Tribe 11 and 25, respectively (*Figure 6A*). The N-terminal sequences of Bs2 and Rx are somewhat similar to MADA sequences but were negative in the HMMER analyses (*Figure 10A*). Interestingly, whereas the N-termini of Bs2 and LOV1 did not complement the cell death activity when swapped into NRC4$^{DV}$, Rx$_{1-17}$ could confer cell death activity when swapped into NRC4$^{DV}$ (*Figure 10C*, *Figure 10—figure supplement 1*). This exception indicates that at least one of the N-terminal sequences that are not predicted as having the MADA motif may still functionally complement the N-terminus of NRC4.

## ZAR1-NRC4 chimeric protein retains the capacity to confer Rpi-blb2-mediated resistance against the late blight pathogen *Phytophthora infestans*

We investigated whether the MADA motif of NRC4 is required for disease resistance against the oomycete pathogen *Phytophthora infestans*. One of the NRC4-dependent sensor NLRs is Rpi-blb2, an NRC-S protein from *Solanum bulbocastanum* that confers resistance to *P. infestans* carrying the matching effector AVRblb2 (*van der Vossen et al., 2003*; *Oh et al., 2009*). For this purpose, we set up a genetic complementation assay in which NRC4 is co-expressed with Rpi-blb2 in leaves of the *N. benthamiana nrc4a/b_9.1.3* mutant prior to inoculation with the *P. infestans* strain 88069 (*Wu et al., 2017*), that carries AVRblb2 (*Figure 11A*). Unlike wild-type NRC4, the NRC4 L9A/V10A/L14A and L9E mutants failed to rescue the resistance to *P. infestans* in the *N. benthamiana nrc4a/b_9.1.3* mutant, indicating that MADA motif mutations not only impair HR cell death as shown above but also affect disease resistance against an oomycete pathogen (*Figure 11B*). We conducted similar complementation assays with the ZAR1$_{1-17}$-NRC4 chimera in which the first 17 amino acids of NRC4 were swapped with the equivalent region of ZAR1, and found that ZAR1$_{1-17}$-NRC4 complemented the *nrc4a/b_9.1.3 N. benthamiana* mutant to a similar degree as wild-type NRC4 (*Figure 11B*). These experiments further confirm that the α1 helix/MADA motif of Arabidopsis ZAR1 is functionally equivalent to the N-terminus of NRC4, and that the chimeric ZAR1$_{1-17}$-NRC4 is not only able to trigger HR cell death but also retains its capacity to function with its NRC-S mate Rpi-blb2 and confer resistance to *P. infestans*.

## Discussion

This study stems from a random truncation screen of the CC-NLR NRC4, which revealed that the very N-terminus of this protein is sufficient to carry out the HR cell death activity of the full-length

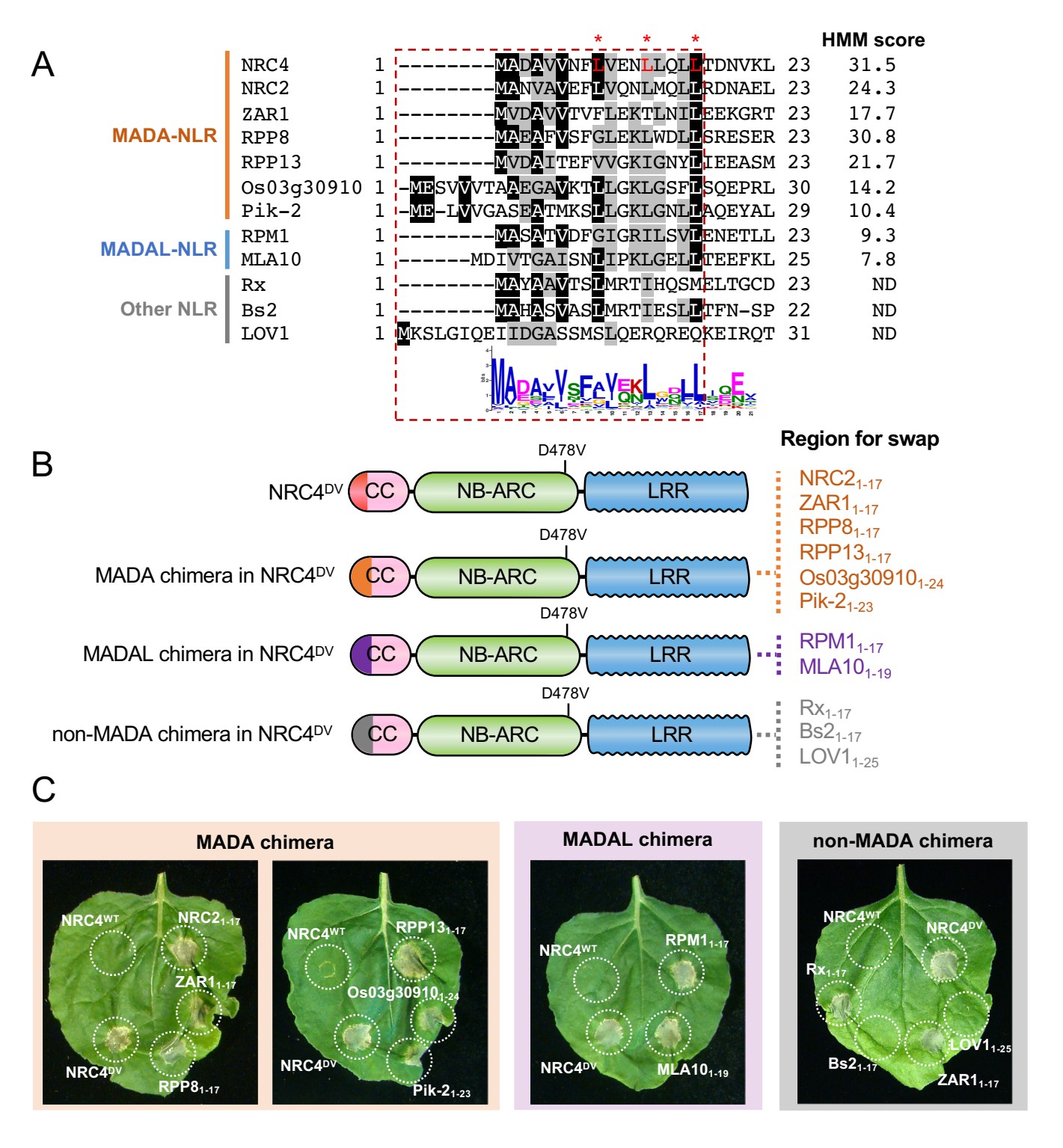

**Figure 10.** First 17 amino acids of NRC4 can be functionally replaced by the N-terminus of other MADA/MADAL-CC-NLRs. (A) Alignment of the N-terminal region of the MADA/MADAL-CC-NLRs. Key residues for cell death activity identified in *Figure 8* are marked as red characters with asterisks in the sequence alignment. Each HMM score is indicated. (B) Schematic representation of NRC4 MADA motif chimeras with MADA, MADAL and non-MADA sequences from other CC-NLRs. The first 17 amino acid region of other MADA-CC-NLR (orange), MADAL-CC-NLR (purple) or non-MADA-CC-NLR (grey) was swapped into NRC4$^{DV}$, resulting in the NRC4 chimeras with MADA/MADAL/non-MADA sequences originated from other NLRs. (C) Cell

*Figure 10 continued on next page*

*Figure 10 continued*
death phenotypes induced by the NRC4 chimeras. NRC4^WT-6xHA, NRC4^DV-6xHA and the chimeras were expressed in *N. benthamiana* leaves. Photographs were taken at 5 days after agroinfiltration.

The online version of this article includes the following figure supplement(s) for figure 10:

**Figure supplement 1.** Quantification of cell death triggered by NRC4 MADA motif chimeras.

protein. It turned out that this region is defined by a consensus sequence—the MADA motif—that occurs in about one fifth of plant CC-NLRs including Arabidopsis ZAR1. The MADA motif covers most of the functionally essential α1 helix of ZAR1 that undergoes a conformational switch during activation of the ZAR1 resistosome (*Wang et al., 2019b*). Our finding that the ZAR1 α1 helix/MADA motif can functionally replace its matching region in NRC4 indicates that the ZAR1 'death switch' mechanism may apply to NRCs and other MADA-CC-NLRs from dicot and monocot plant species.

We recently proposed that NLRs may have evolved from multifunctional singleton receptors to functionally specialized and diversified receptor pairs and networks (*Adachi et al., 2019a*). In this study, a striking finding from the computational analyses is that all NRC-S lack the MADA motif even though they are more closely related to NRC-H than to ZAR1 and other MADA-CC-NLRs in the NB-ARC phylogenetic tree (*Figure 6*). These observations led us to draw the evolutionary model of *Figure 12*. In this model, we propose that MADA-type sequences have emerged early in the evolution of CC-NLRs and have remained conserved from singletons to helpers in NLR pair and network throughout evolution. In sharp contrast, MADA sequences appear to have degenerated over time in sensor CC-NLRs as these proteins specialized in pathogen detection and lost the capacity to execute the immune response without their helper mates. Consistent with this view, NRC-H are known to be more highly conserved than their NRC-S partners within the Solanaceae (*Wu et al., 2017*; *Stam et al., 2019*). Future analyses will determine whether MADA-CC-NLRs are generally more evolutionarily constrained than non-MADA containing NLRs.

In addition, about half of the NRC-S proteins have acquired N-terminal extensions (N-terminal domains) before their CC domain, which would preclude a free N-terminal α1 helix essential for a ZAR1 type 'death switch' mechanism (*Figure 6*). In fact, the N-terminal domains of Prf and Sw5b function as baits that sense pathogen effectors, suggesting functional analogy to integrated effector detection motifs found in some NLRs, and are not known to be involved in executing the immune response (*Saur et al., 2015*; *Li et al., 2019a*). Here, we hypothesize that the CC domains of these and other sensor NLRs have extensively diversified over evolutionary time and are losing the capacity to function as HR cell death executors. This could be a consequence of relaxed selection given that these proteins rely on their MADA-CC-NLR partners to execute the immune response as discussed above. Additional structure-function experiments will be needed to determine the extent to which this 'use-it-or-lose-it' evolutionary model applies to the sensor sub-class of NLR immune receptors.

Understanding the precise nature of the N-terminal sequences that can functionally replace the α1 helix requires further investigation. In the MADA motif swap experiments, we found one exception to the correlation between MADA predictions and functional complementation of NRC4. The N-terminal sequence of the NRC-S NLR Rx, which was negative in the MADA HMMER searches, complemented the cell death activity of NRC4 MADA motif (*Figure 10*, *Figure 10—figure supplement 1*). Nonetheless, previously the NB domain of Rx was reported to be capable of triggering cell death (*Rairdan et al., 2008*), suggesting that the CC domain of Rx is dispensable for activation of HR. Therefore, in our 'use-it-or-lose-it' model, the N-termini of some NRC-S may not have fully degenerated into non-functional sequences and may have residual ability to functionally complement MADA. In the future, it would be fascinating to determine resistosome configurations of NLR sensor and helper hetero-complexes. As discussed elsewhere (*Adachi et al., 2019a*; *Jubic et al., 2019*), one hypothesis is that sensor NLRs associate with the resistosome as functional equivalents of RLCKs in the ZAR1 resistosome. Another is that sensor NLRs form one of the wheel spokes in a hetero-oligomeric resistosome as in the mammalian NAIP/NLRC4 inflammasome (*Tenthorey et al., 2017*). It is possible that in this configuration, the N-terminus of a sensor NLR such as Rx remains evolutionarily constrained in terms of length and sequence composition. Future structural analyses of NLR sensor/helper heterocomplexes are needed to address these questions.

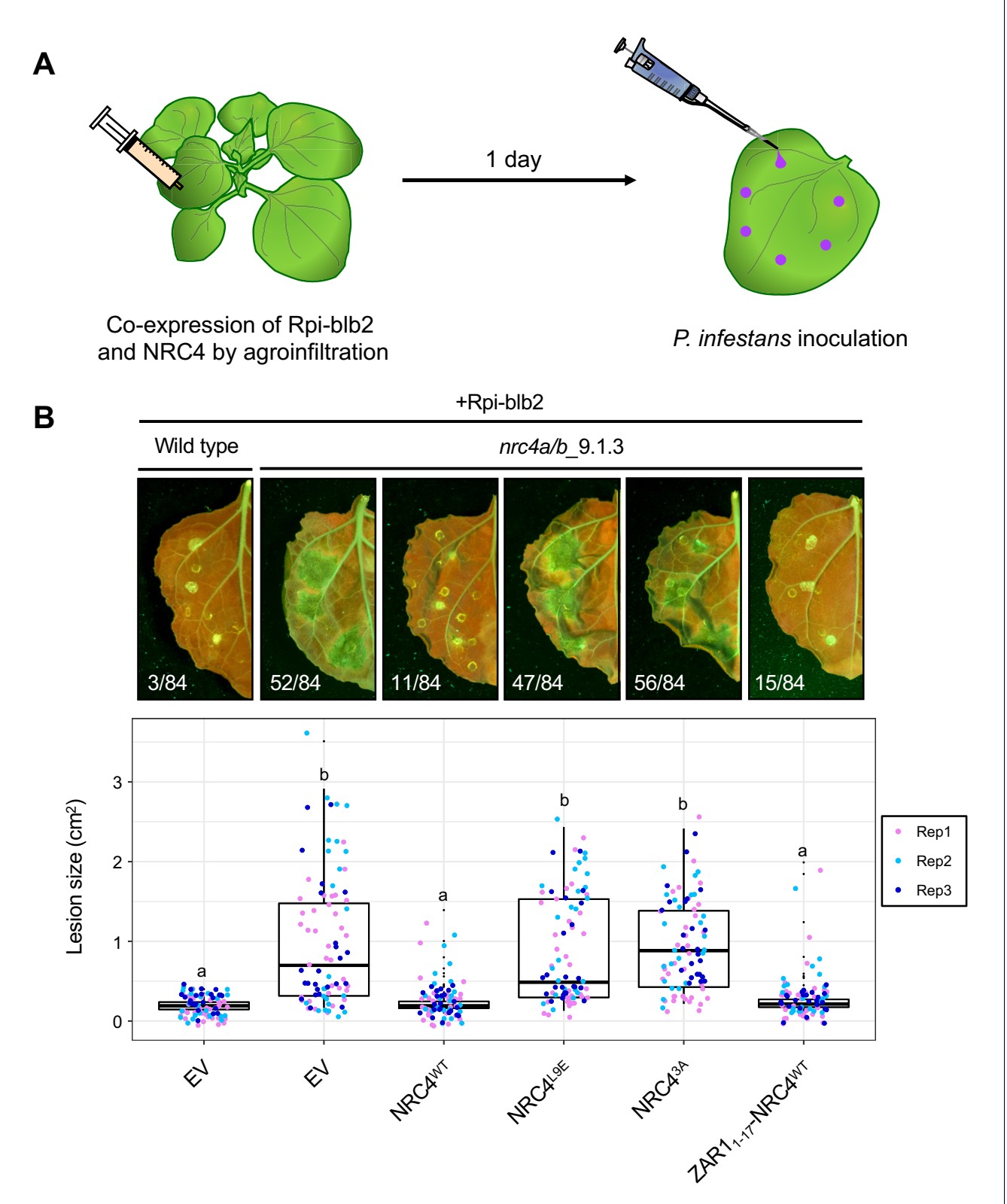

**Figure 11.** The chimeric protein ZAR1$_{1-17}$-NRC4 complements NRC4 function in Rpi-blb2-mediated resistance. (**A**) Schematic representation of NRC4 complementation assay for Rpi-blb2-mediated resistance. Wild-type and the variants of NRC4 were co-expressed with RFP-Rpiblb2 in wild-type or *nrc4a/b*_9.1.3 *N. benthamiana* leaves by agroinfiltration. The leaves were inoculated with droplets of zoospore suspension from *P. infestans* strain 88069 at 1 day after the agroinfiltration. The syringe and pipet are not drawn to scale. (**B**) Disease and resistance phenotypes on NRC4/Rpi-blb2-expressed

*Figure 11 continued on next page*

*Figure 11 continued*

leaves. Images were taken under UV light at 7 days post inoculation. The lesion size (bottom panel) was measured in Fiji (Fiji Is Just ImageJ). Experiments were repeated three times with totally 84 inoculation site each. The numbers on the photographs indicate the sum of spreading lesions/ total inoculation sites from the three replicates. Statistical differences among the samples were analysed with Tukey's HSD test (p<0.01).

Already, our evolutionary model appears to be consistent with some paired NLR configurations in addition to the NRC-H/NRC-S network. One example is rice Pik-1 and Pik-2, which are a well-established NLR pair that detects the AVRPik effector of the rice blast fungus *M. oryzae* (*Maqbool et al., 2015*; *Białas et al., 2018*). AVRPik binding to the integrated heavy metal associated (HMA) domain of Pik-1 results in HR cell death and blast fungus resistance only in the presence of its helper Pik-2 protein (*Maqbool et al., 2015*). In our computational analyses only Pik-2 was detected to carry an N-terminal MADA motif (*Figure 5*, HMM score = 10.4) even though the CC domains of both proteins grouped into Tribe 1 (*Figure 3*). The Pik-2 MADA motif could substitute for the N-terminus of NRC4 in our cell death assays despite having six additional amino-acids at its N-terminus (*Figure 10*). These results are consistent with our *Figure 11* model and imply that the helper NLR Pik-2 may execute HR cell death via its N-terminal MADA motif whereas its paired sensor NLR Pik-1 does not have the capacity to carry this activity on its own.

In addition to ZAR1, RPP8 is another Arabidopsis MADA-CC-NLR with high similarity to the N-terminus of NRC4 with nine invariant amino acids out of 17 (53%; HMMER score = 30.8). This RPP8 MADA motif could substitute for the N-terminus of NRC4 indicating that it is functional (*Figure 10*). In Arabidopsis, RPP8 (AT5G43470) and its paralogs occur at dynamic genetic loci that exhibit frequent sequence exchanges as deduced from comparative genomic analyses (*Kuang et al., 2008*). Four of the five RPP8 paralogs in the Arabidopsis ecotype Col-0 were deemed to have a MADA motif based on our HMMER searches, whereas a fifth paralog LOV1 (AT1G10920) was negative and did not complement NRC4 autoactivity in the MADA motif swap experiments (*Figure 10*, *Figure 10—figure supplement 1*). LOV1 confers sensitivity to the victorin effector produced by the necrotrophic fungus *Cochliobolus victoriae* by binding the defense-associated thioredoxin TRX-h5

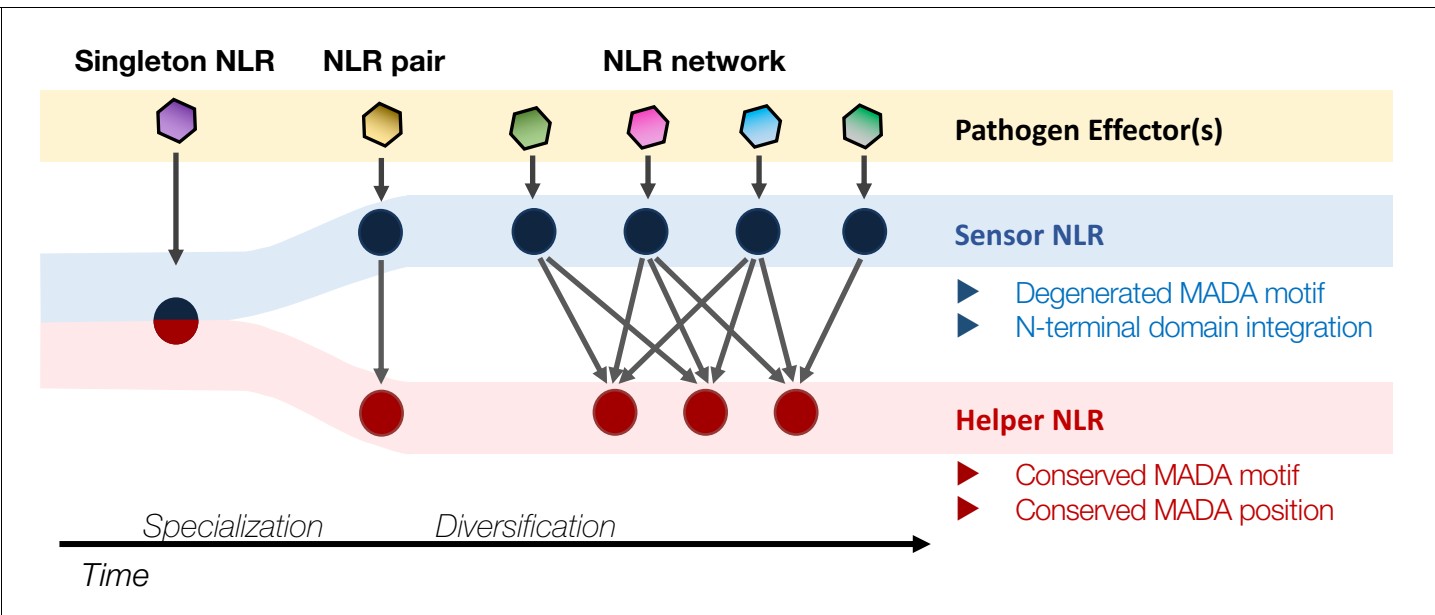

**Figure 12.** Evolution of NLRs from singletons to networks. We propose that the N-terminal MADA motif/α1 helix has emerged early in the evolution of CC-NLRs and has remained constrained throughout time as singletons evolved from multifunctional proteins into specialized paired and networked NLR helpers. In contrast, the MADA motif/α1 helix has degenerated in sensor CC-NLRs as they rely on their NLR helper mates for executing the immune response ('use-it-or-lose-it' model of evolution). In addition, some sensor NLRs, such as a large subset of NRC-S proteins, have acquired N-terminal domains (NTDs)—prior to their CC domains—that function in pathogen detection. Such NTDs would preclude a free N-terminal α1 helix, which would be incompatible with the current model of ZAR1 resistosome activation.

when it is complexed with victorin (*Lorang et al., 2012*). Interestingly, LOV1 binds TRX-h5 via its CC domain indicating that this region has evolved a pathogen sensor activity in this NLR protein (*Lorang et al., 2012*). How the sensor activity of the CC domain of LOV1 relates to the absence of a detectable MADA motif and whether this protein relies on other MADA-CC-NLRs to execute the cell death response are unanswered questions that are raised by these observations.

In activated ZAR1 resistosome, a funnel-shaped structure formed by five α1 helices is thought to directly execute hypersensitive cell death by forming a toxin-like pore in the plasma membrane (*Wang et al., 2019b*). To what extent do activated MADA-CC-NLRs function according to this ZAR1 model? Structure informed mutagenesis of ZAR1 revealed that F9, L10 and L14 on the outer surface of the funnel-shaped structure are required for immunity (*Wang et al., 2019b*). Here, our Ala and Glu scans of the MADA motif revealed that the NRC4 L9, L13 and L17 residues are essential for HR cell death activity. All three residues mapped to the outer surface of NRC4 α1 helices as predicted from a homology model based on the ZAR1 resistosome (*Figure 8—figure supplement 2*). We also found that mutations that perturb the MADA motif and prevent YFP self-association impair the capacity of NRC4$_{1-29}$-YFP to cause cell death and form puncta in *N. benthamiana* leaf cells (*Figure 2—figure supplement 2*, *Figure 7—figure supplement 1*, *Figure 9*). Our current interpretation of these results is that NRC4$_{1-29}$-YFP forms high-order complexes to cause cell death. However, direct support for this hypothesis is still missing. In addition, we lack detailed analyses of the cellular dynamics of the NRC4$_{1-29}$-YFP puncta and the degree to which they associate with membrane compartments in living plant cells. In the future, further biochemical, structural and cellular analyses are needed to determine the precise nature of the broadly conserved MADA motif and address the extent to which the ZAR1 'death switch' model occurs in CC-NLRs.

As discussed by *Wang et al. (2019b)*, the interior space of the funnel structure is also important because the ZAR1 double mutant E11A/E18A is impaired in cell death and disease resistance activities. However, in our Glu mutant scan, we failed to observe a reduction in HR cell death activities with single site mutants in these residues or other amino acids that are predicted to line up the interior space of the funnel-shaped structure. Whether or not this reflects genuine biological differences between ZAR1 and NRC4 remains to be studied.

A subset of CC-NLRs of the RPW8/HR family of atypical resistance proteins have a distinct type of coiled-coil domain known as CC$_R$ (*Barragan et al., 2019*; *Li et al., 2019b*). We failed to detect any MADA type sequences in these CC$_R$-NLR proteins. Indeed, the CC$_R$ domain has similarity to mixed lineage kinase domain-like (MLKL) proteins and fungal HeLo/HELL domains, which form multi-helix bundles and act as membrane pore forming toxins (*Barragan et al., 2019*; *Li et al., 2019a*; *Mahdi et al., 2019*). Whether the CC$_R$ domains function as a distinct cell death inducing system in plants compared to MADA-CC-NLRs remains to be determined. Interestingly, Arabidopsis HR4, a CC$_R$ containing protein, interacts in an allele-specific manner with the genetically unlinked CC-NLR RPP7b to trigger autoimmunity in the absence of pathogens (*Barragan et al., 2019*). Recently, *Li et al. (2019b)* showed that RPP7b forms higher-order complexes of six to seven subunits only when activated by the matching autoimmune HR4$^{Fei-0}$ protein in a biochemical process reminiscent of activated ZAR1 resistosome (*Li et al., 2019a*). In our HMMER searches, RPP7b and its four Arabidopsis paralogs were all classed as carrying the MADA motif. Thus, findings by *Li et al. (2019b)* directly link a MADA-CC-NLR to the formation of resistosome type structures consistent with our view that the ZAR1 model widely applies to other NLRs with the MADA α1 helix. It will be fascinating to determine whether or not RPP7b and HR4 are both capable of executing cell death, especially as two-component systems of NLR and HeLo/HELL proteins are common in fungi and mammals (*Barragan et al., 2019*).

Plant NLRs can be functionally categorized into singleton, sensor or helper NLRs based on their biological activities (*Adachi et al., 2019a*). However, it remains challenging to predict NLR functions from the wealth of unclassified NLRomes that are emerging from plant genome sequences. It has not escaped our attention that the discovery of the MADA motif as a signature of NLR singletons and helpers—but missing in sensor NLRs—enables the development of computational pipelines for predicting NLR networks from naïve plant genomes. Such in silico predictions can be tested by co-expression of paired NLRs in *N. benthamiana*. In addition, MADA motif predictions can be validated using our straightforward functional assay of swapping the NRC4 N-terminus, with the readouts consisting of both HR cell death (*Figure 10*) and resistance to *P. infestans* (*Figure 11*). Dissecting the NLR network architecture of plant species is not only useful for basic mechanistic studies but has

also direct implications for breeding disease resistance into crop plants and reducing the autoimmune load of NLRs (*Chae et al., 2016*; *Wu et al., 2018*; *Adachi et al., 2019a*).

# Materials and methods

## Key resources table

| Reagent type (species) or resource | Designation | Source or reference | Identifiers | Additional information |
|---|---|---|---|---|
| Genetic reagent (*Nicotiana benthamiana*) | NRC4-KO *N. benthamiana* (*nrc4a/b_9.1.3* and *nrc4a/b_1.2.1*) | This paper | | Materials and methods: Generation of *N. benthamiana nrc4a/b* CRISPR/Cas9 mutants |
| Recombinant DNA reagent | pGEM::Mu-STOP | This paper | | Materials and methods: Mu-STOP in vitro transposition |
| Commercial assay, kit | Mutation Generation System Kit | Thermo Fisher | Cat #: F-701 | Materials and methods: Mu-STOP in vitro transposition |
| Gene (*Solanum lycopersicum*) | Tomato genome sequence (Tomato ITAG release 2.40) | Sol Genomics Network (https://solgenomics.net/) | | Materials and methods: Bioinformatic and phylogenetic analyses |
| Gene (*N. benthamiana*) | *N. benthamiana* genome sequence (*N. benthamiana* Genome v0.4.4) | Sol Genomics Network (https://solgenomics.net/) | | Materials and methods: Bioinformatic and phylogenetic analyses |
| Gene (*Arabidopsis thaliana*) | Arabidopsis genome sequence (Araport11) | https://www.araport.org/ | | Materials and methods: Bioinformatic and phylogenetic analyses |
| Gene (*Beta vulgaris*) | Sugar beet genome sequence (RefBeet-1.2) | http://bvseq.molgen.mpg.de/index.shtml | | Materials and methods: Bioinformatic and phylogenetic analyses |
| Gene (*Oryza sativa*) | Rice genome sequence (Rice Gene Models in Release 7) | http://rice.plantbiology.msu.edu/ | | Materials and methods: Bioinformatic and phylogenetic analyses |
| Gene (*Hordeum vulgare*) | Barley genome sequence (IBSC_v2) | https://www.barleygenome.org.uk/ | | Materials and methods: Bioinformatic and phylogenetic analyses |
| Other | 3D structure of ZAR1 | Protein Data Bank | 6J5T | Materials and methods: Structure homology modelling |

## Plant growth conditions

Wild type and mutant *N. benthamiana* were propagated in a glasshouse and, for most experiments, were grown in a controlled growth chamber with temperature 22–25°C, humidity 45–65% and 16/8 hr light/dark cycle.

## Generation of *N. benthamiana nrc4a/b* CRISPR/Cas9 mutants

Constructs for generating *NRC4* knockout *N. benthamiana* were assembled using the Golden Gate cloning method (*Weber et al., 2011*; *Nekrasov et al., 2013*; *Belhaj et al., 2013*). sgRNA4.1 and sgRNA4.2 were cloned under the control of the Arabidopsis (*Arabidopsis thaliana*) U6 promoter (AtU6pro) [pICSL90002, The Sainsbury Laboratory (TSL) SynBio] and assembled in pICH47751 (Addgene no. 48002) and pICH47761 (Addgene no. 48003), respectively as previously described (*Belhaj et al., 2013*). Primers sgNbNRC4.1_F (tgtggtctcaATTGAAAAACGGTACATACCGCAGgttttagagctagaaatagcaag), sgNbNRC4.2_F (tgtggtctcaATTGAGTCAGGAATCTTGCAGCTGgttttagagctagaaatagcaag) and sgRNA_R (tgtggtctcaAGCGTAATGCCAACTTTGTAC) were used to clone sgRNA4.1 and sgRNA4.2. pICSL11017::NOSpro::BAR (TSL SynBio), pICSL11021::35Spro::Cas9

(Addgene no. 49771), pICH47751::AtU6p::sgRNA4.1, pICH47761::AtU6pro::sgRNA4.2, and the linker pICH41780 (Addgene no. 48019) were assembled into the vector pICSL4723 (Addgene no. 48015) as described (*Weber et al., 2011*) resulting in construct pICSL4723::BAR::Cas9::sgRNA4.1::sgRNA4.2 that was used for plant transformation. Transgenic *N. benthamiana* were generated by TSL Plant Transformation team as described before (*Nekrasov et al., 2013*).

## *N. benthamiana nrc4a/b* genotyping

Genomic DNA of selected T2 *N. benthamiana* transgenic plants *nrc4a/b*_9.1.3 and *nrc4a/b*_1.2.1 was extracted using DNeasy Plant DNA Extraction Kit (Qiagen). Primers NRC4_1_F (GGAAG TGCAAAGGGAGAGTT), NRC4_1_R (TCGCCTGAACCACAAACTTA), NRC4_2_F (GGCAAGAA TTTTGGATGTGG) and NRC4_2_R (CGAGGAACCCTTTTTAGGCAG) were used in multiplex polymerase chain reaction (PCR) assays to amplify the region targeted by the two sgRNAs. Multiplex amplicon sequencing was performed by the Hi-Plex technique (*Lyon et al., 2016*). Sequence reads were aligned to the reference *N. benthamiana* draft genome Niben.genome.v0.4.4 [Sol Genomics Network (SGN), https://solgenomics.net/], and *NRC4a* (on scaffold Niben044Scf00002971) and *NRC4b* (on scaffold Niben044Scf00016103) were further analysed. T3 lines from the selected T2 plants were used for the experiments.

## Plasmid constructions

To generate NRC4$_{1-29}$-YFP expression construct, NRC4$_{1-29}$ coding sequence was amplified by Phusion High-Fidelity DNA Polymerase (Thermo Fisher), and the purified amplicon was directly used in Golden Gate assembly with pICH85281 [mannopine synthase promoter+Ω (MasΩpro), Addgene no. 50272], pICSL50005 (YFP, TSL SynBio), pICSL60008 [Arabidopsis heat shock protein terminator (HSPter), TSL SynBio] into binary vector pICH47742 (Addgene no. 48001). Primers used for NRC4$_{1-29}$ coding sequences are listed in *Supplementary file 1*.

To generate an autoactive mutant of *N. benthamiana* NRC4, the aspartic acid (D) in the MHD motif was substituted to valine (V) by site-directed mutagenesis using Phusion High-Fidelity DNA Polymerase (Thermo Fisher). pCR8::NRC4$^{WT}$ (*Wu et al., 2017*) was used as a template. Primers NRC4_D478V_F (5'-Phos/ATGTTGCATCAGTTCTGCAAAAAGGAGGCT) and NRC4_D478V_R (5'-Phos/GACGTGAAGACGACATGTTTTTATTTGACC) were used for introducing the mutation in the PCR. The mutated NRC4 was verified by DNA sequencing of the obtained plasmid.

pCR8::NRC4$^{WT}$ (*Wu et al., 2017*) or pCR8::NRC4$^{DV}$ without its stop codon were used as a level 0 modules for the following Golden Gate cloning. NRC4$^{DV}$-3xFLAG was generated by Golden Gate assembly with pICH51266 [35S promoter+Ω promoter, Addgene no. 50267], pICSL50007 (3xFLAG, Addgene no. 50308) and pICH41432 (octopine synthase terminator, Addgene no. 50343) into binary vector pICH47732 (Addgene no. 48000). NRC4$^{WT}$-6xHA and NRC4$^{DV}$-6xHA were generated by Golden Gate assembly with pICH85281 (MasΩpro), pICSL50009 (6xHA, Addgene no. 50309), pICSL60008 (HSPter) into the binary vector pICH47742. NRC4$^{WT}$-YFP and NRC4$^{DV}$-YFP were generated by Golden Gate assembly with pICH85281 (MasΩpro), pICSL50005 (YFP), pICSL60008 (HSPter) into binary vector pICH47742. For free YFP expression construct, pAGM3212 (YFP, TSL SynBio) was assembled with pICH85281 (MasΩpro) and pICSL60008 (HSPter) into the binary vector pICH47742 by Golden Gate reaction.

To reduce homo-affinity of YFP, YFP alanine (A) 206 was substituted to lysine (K) (*Zacharias et al., 2002*), by site-directed mutagenesis using Phusion High-Fidelity DNA Polymerase (Thermo Fisher). pAGM3212 (YFP, TSL SynBio) was used as a template. Primers used for mutagenesis are listed in *Supplementary file 1*. The amplicons were directly used in Golden Gate assembly with pICH41308 (Addgene no. 47998) or pAGM1301 (Addgene no. 47989). pICH41308::YFP$^{A206K}$ was assembled with pICH85281 (MasΩpro) and pICSL60008 (HSPter) into the binary vector pICH47742 by Golden Gate reaction. pAGM1301::YFP$^{A206K}$ was assembled with pCR8::NRC4$^{DV}$ or NRC4$_{1-29}$ amplicon, pICH85281 (MasΩpro) and pICSL60008 (HSPter) into the binary vector pICH47742 by Golden Gate reaction.

To generate MADA motif mutants and chimeras of NRC4, the full-length sequence of NRC4$^{WT}$ or NRC4$^{DV}$ was amplified by Phusion High-Fidelity DNA Polymerase (Thermo Fisher) with the forward primers listed in *Supplementary file 1*. Purified amplicons were cloned into pCR8/GW/D-TOPO (Invitrogen) as a level 0 module. The level 0 plasmids were then used for Golden Gate assembly with

pICH85281 (MasΩpro), pICSL50009 (6xHA) and pICSL60008 (HSPter) into the binary vector pICH47742.

To generate pTRBO::YFP, pTRBO::ZAR1$_{1-144}$-YFP, pTRBO::ZAR1$_{1-144}$$^{F9A/L10A/L14A}$-YFP, pTRBO::NRC4$_{1-29}$-YFP and pTRBO::NRC4$_{1-29}$$^{L9A/V10A/L14A}$-YFP plasmids, we used GENEWIZ Standard Gene Synthesis with custom vector cloning service into the pTRBO vector (*Lindbo, 2007a*).

## Mu-STOP in vitro transposition

To generate the Mu-STOP transposon (*Poussu, 2005*), entranceposon M1-KanR (Mutation Generation System Kit, Thermo Fisher) was used as a PCR template, and three translational stop signals were added to each transposon end by Phusion High-Fidelity DNA Polymerase and Mu-STOP primer (GGAAGATCTGATTGATTGAACGAAAAACGCGAAAGCGTTTC). The 3' A overhang was then introduced to the Mu-STOP amplicon by DreamTaq DNA polymerase (Thermo Fisher), and the resulting Mu-STOP amplicon was cloned into pGEM-T Easy (Promega). Mu-STOP transposon was then released from pGEM::Mu-STOP by *Bgl*II digestion and purified by GeneJET Gel Extraction Kit (Thermo Fisher). 100 ng of the purified Mu-STOP transposon was mixed with 500 ng of the target plasmid, pICH47732::35SΩpro::NRC4$^{DV}$-3xFLAG, and MuA transposase from the Mutation Generation System Kit (Thermo Fisher). The in vitro transposition reaction was performed according to the manufacturer's procedure and carried out at 30°C for 6 hr.

The NRC4$^{DV}$::Mu-STOP library was transformed into *Agrobacterium tumefaciens* Gv3101 by electroporation. Mu-STOP insertion sites were determined by colony PCR using DreamTaq DNA polymerase (Thermo Fisher) and PCR amplicon sequencing. For the PCR, we used a forward primer (GAACCCTGTGGTTGGCATGCACATAC) matching pICH47732 and a reverse primer (CAACGTGGC TTACTAGGATC) matching Mu-STOP transposon.

## Transient gene-expression and cell death assays

Transient expression of NRC wild-type and mutants, as well as other genes, in *N. benthamiana* were performed by agroinfiltration according to methods described by *Bos et al. (2006)*. Briefly, four-weeks old *N. benthamiana* plants were infiltrated with *A. tumefaciens* strains carrying the binary expression plasmids. *A. tumefaciens* suspensions were prepared in infiltration buffer (10 mM MES, 10 mM MgCl$_2$, and 150 µM acetosyringone, pH5.6) and were adjusted to OD$_{600}$ = 0.5. For transient expression of NRC4$^{WT}$-YFP, NRC4$^{DV}$-YFP, NRC4$_{1-29}$-YFP, free YFP and the YFP$^{A206K}$ variants, the *A. tumefaciens* suspensions (OD$_{600}$ = 0.25) were mixed in a 1:1 ratio with an *A. tumefaciens* expressing p19, the suppressor of posttranscriptional gene silencing of *Tomato bushy stunt virus* that is known to enhance in planta protein expression (*Lindbo, 2007b*). HR cell death phenotypes were scored according to the scale of *Segretin et al. (2014)* modified to range from 0 (no visible necrosis) to 7 (fully confluent necrosis). In *Figure 2—figure supplement 2* and *Figure 7—figure supplement 1*, cell death was visualized with Odyssey Infrared Imager (800 nm channel, LI-COR).

## Protein immunoblotting

Protein samples were prepared from six discs (8 mm diameter) cut out of *N. benthamiana* leaves at 1 day after agroinfiltration and were homogenised in extraction buffer [10% glycerol, 25 mM Tris-HCl, pH 7.5, 1 mM EDTA, 150 mM NaCl, 2% (w/v) PVPP, 10 mM DTT, 1x protease inhibitor cocktail (SIGMA), 0.2% IGEPAL (SIGMA)]. The supernatant obtained after centrifugation at 12,000 x*g* for 10 min was used for SDS-PAGE. Immunoblotting was performed with HA-probe (F-7) HRP (Santa Cruz Biotech) or anti-GFP antibody (ab290, abcam) in a 1:5000 dilution. Equal loading was checked by taking images of the stained PVDF membranes with Pierce Reversible Protein Stain Kit (#24585, Thermo Fisher).

## Bioinformatic and phylogenetic analyses

We used NLR-parser (*Steuernagel et al., 2015*) to identify NLR sequences from the protein databases of tomato (SGN, Tomato ITAG release 2.40), *N. benthamiana* (SGN, *N. benthamiana* Genome v0.4.4), Arabidopsis (https://www.araport.org/, Araport11), sugar beet (http://bvseq.molgen.mpg. de/index.shtml, RefBeet-1.2), rice (http://rice.plantbiology.msu.edu/, Rice Gene Models in Release 7) and barley (https://www.barleygenome.org.uk/, IBSC_v2). The obtained NLR sequences, from NLR-parser, were aligned using MAFFT v. 7 (*Katoh and Standley, 2013*), and the protein sequences that

lacked the p-loop motif were discarded from the NLR dataset. The gaps in the alignments were deleted manually in MEGA7 (*Kumar et al., 2016*) and the NB-ARC domains were used for generating phylogenetic trees (*Figure 3—figure supplement 1—source data 1*). The neighbour-joining tree was made using MEGA7 with JTT model and bootstrap values based on 100 iterations (*Figure 3—figure supplement 1*). We removed TIR-NLR clade members from the final database, and retained all CC-NLR sequences, including the $CC_R$-NLR (RPW8-NLR), that possess N-terminal domains longer than 30 amino acids (988 protein sequences, *Figure 3—source data 1*).

The NB-ARC domain sequences from 988 proteins (*Figure 3—figure supplement 2—source data 2*) were used to construct the CC-NLR phylogenetic tree in *Figure 3—figure supplement 2*. The neighbour-joining tree was constructed as described above.

For the tribe analyses, we extracted the N-terminal domain sequences, prior to NB-ARC domain, from the CC-NLR database (*Figure 3—source data 2*), and used the Tribe-MCL feature from Markov Cluster Algorithm (*Enright et al., 2002*) to cluster the sequences into tribes with BLASTP E-value cutoff $<10^{-8}$. NLRs in each tribe were subjected to motif searches using the MEME (Multiple EM for Motif Elicitation) v. 5.0.5 (*Bailey and Elkan, 1994*) with parameters 'zero or one occurrence per sequence, top five motifs', to detect consensus motifs conserved in $\geq 70\%$ of input sequences.

We used the most N-terminal motif detected in Tribe 2 from the MEME analysis to construct a hidden Markov model (HMM) for the MADA motif. Sequences aligned to the MADA motif were extracted in Stockholm format and used in hmmbuild program implemented in HMMER v2.3.2 (*Eddy, 1998*). The HMM was then calibrated with hmmcalibrate. This MADA-HMM (*Supplementary file 2*) was used to search the CC-NLR database (*Figure 3—source data 1*) with the hmmsearch program (hmmsearch –max -o < outputfile > <hmmfile > <seqdb > ). To estimate the false positive rate, hmmsearch program was applied to full Arabidopsis and tomato proteomes (Araport11 and ITAG3.2) with the MADA-HMM and the output is displayed in *Figure 4—source data 1* and discussed in the results section.

## Pathogen infection assays

*P. infestans* infection assays were performed by applying droplets of zoospore suspension on detached leaves as described previously (*Song et al., 2009*). Briefly, leaves of five-weeks old wild-type and *nrc4a/b N. benthamiana* plants were infiltrated with *A. tumefaciens* solutions, in which each *Agrobacterium* containing a plasmid expressing RFP::Rpi-blb2 (*Wu et al., 2017*) was mixed in a 1:1 ratio ($OD_{600}$ = 0.5 for each strain) with *Agrobacterium* containing either the empty vector, wild type NRC4, or NRC4 variant. At 24 hr after agroinfiltration, the abaxial side of the leaves were inoculated with 10 µL zoospore suspension (100 zoospores/µL) of *P. infestans* strain 88069 prepared according to the methods reported by *Song et al. (2009)*. The inoculated leaves were kept in a moist chamber at room temperature (21–24°C) for 7 days, and imaged under UV light (UVP Blak-Ray B-100AP lights – 365 nm) with Wratten No.8 Yellow Filter for visualization of the lesions. The camera setting was ISO 1600, White Balance 6250K, F11 and 10 s exposure.

## Structure homology modelling

We used the cryo-EM structure of activated ZAR1 (*Wang et al., 2019b*) as template to generate a homology model of NRC4. The amino acid sequence of NRC4 was submitted to Protein Homology Recognition Engine V2.0 (Phyre2) for modelling (Kelley et a., 2015). The coordinates of ZAR1 structure (6J5T) were retrieved from the Protein Data Bank and assigned as modelling template by using Phyre2 Expert Mode. The resulting model of NRC4 comprised amino acids Val-5 to Glu-843 and was illustrated in CCP4MG software (*McNicholas et al., 2011*).

## Microscopy

For localization analyses, leaf discs (6 mm in diameter) of *N. benthamiana* leaves were made 2 days after agroinfiltration and were used for imaging. Images were captured with Leica SP8 resonant inverted confocal microscope (Leica Microsystems). For excitation, Argon laser and Helium-Neon laser wer set to 514 nm and 633 nm, respectively. Hybrid detectors were used with 517–575 and 584–638 nm bandpass filters to capture YFP and RFP signals, respectively. Gain, laser intensities and zoom were kept the same for all images. Images were processed in FIJI (Fiji Is Just ImageJ).

## Accession numbers

The NRC4 sequences used in this study can be found in the Solanaceae Genomics Network (SGN) or GenBank/EMBL databases with the following accession numbers: NbNRC4 (NbNRC4, MK692737; NbNRC4a, Niben044Scf00002971; NbNRC4b, Niben044Scf00016103).

## Acknowledgements

We are thankful to several colleagues for discussions and ideas. We thank Matthew Smoker and other members of the TSL Plant Transformation facility as well as Mark Youles of TSL SynBio for invaluable technical support. We thank Kurt Lamour for advice on genotyping the CRISPR/Cas9 mutants. HA is funded by the Japan Society for the Promotion of Science (JSPS) and LD by a Marie Sklodowska-Curie Actions (MSCA) Fellowship. The Kamoun Lab is funded primarily from the Gatsby Charitable Foundation, Biotechnology and Biological Sciences Research Council (BBSRC, UK), and European Research Council (ERC; NGRB and BLASTOFF projects).

## Additional information

### Competing interests

Chih-hang Wu: SK, LD and CH-W filed a patent on NRCs.(WO/2019/108619). Lida Derevnina: SK, LD and CH-W filed a patent on NRCs. (WO/2019/108619). Sophien Kamoun: SK, LD and CH-W filed a patent on NRCs. SK receives funding from industry on NLR biology.(WO/2019/108619). The other authors declare that no competing interests exist.

### Funding

| Funder | Author |
| --- | --- |
| Gatsby Charitable Foundation | Sophien Kamoun |
| Biotechnology and Biological Sciences Research Council | Sophien Kamoun |
| European Research Council | Sophien Kamoun |

The funders had no role in study design, data collection and interpretation, or the decision to submit the work for publication.

### Author contributions

Hiroaki Adachi, Conceptualization, Resources, Data curation, Formal analysis, Validation, Investigation, Visualization, Methodology; Mauricio P Contreras, Adeline Harant, Chih-hang Wu, Toshiyuki Sakai, Data curation, Formal analysis, Investigation; Lida Derevnina, Data curation, Funding acquisition, Investigation; Cian Duggan, Eleonora Moratto, Validation; Tolga O Bozkurt, Abbas Maqbool, Joe Win, Supervision; Sophien Kamoun, Conceptualization, Data curation, Supervision, Funding acquisition, Visualization, Project administration

### Author ORCIDs

Hiroaki Adachi http://orcid.org/0000-0002-7184-744X
Mauricio P Contreras https://orcid.org/0000-0001-6002-0730
Cian Duggan http://orcid.org/0000-0001-7302-7472
Tolga O Bozkurt http://orcid.org/0000-0003-0507-6875
Joe Win https://orcid.org/0000-0002-9851-2404
Sophien Kamoun https://orcid.org/0000-0002-0290-0315

### Decision letter and Author response

Decision letter https://doi.org/10.7554/eLife.49956.sa1
Author response https://doi.org/10.7554/eLife.49956.sa2

# Additional files

## Supplementary files

• Supplementary file 1. Primers used for generating NRC4 variants by Golden Gate cloning.

• Supplementary file 2. The MADA-HMM for HMMER analysis. This MADA-HMM was used for searching MADA-CC-NLRs from CC-NLR database (*Figure 3—source data 1*).

• Transparent reporting form

## Data availability

All sequence data used for bioinformatic and phylogenetic analyses are included in the manuscript and supporting files.

The following previously published datasets were used:

| Author(s) | Year | Dataset title | Dataset URL | Database and Identifier |
|---|---|---|---|---|
| Fernandez-Pozo N, Menda N, Edwards JD, Saha S, Tecle IY, Strickler SR, Bombarely A, Fisher-York T, Pujar A, Foerster H, Yan A, Mueller LA | 2015 | The Sol Genomics Network (SGN)– from genotype to phenotype to breeding | ftp://ftp.solgenomics.net/genomes/Nicotiana_benthamiana/assemblies/ | Sol Genomics Network, N. benthamiana Genome v0.4.4 |
| Cheng CY, Krishnakumar V, Chan AP, Thibaud-Nissen F, Schobel S, Town CD | 2017 | Araport11: a complete reannotation of the Arabidopsis thaliana reference genome | ftp://ftp.ncbi.nlm.nih.gov/genomes/all/GCA/000/001/735/GCA_000001735.2_TAIR10.1 | NCBI genome, Araport11 |
| Dohm JC, Minoche AE, Holtgräwe D, Capella-Gutiérrez S, Zakrzewski F, Tafer H, Rupp O, Sörensen TR, Stracke R, Reinhardt R, Goesmann A, Kraft T, Schulz B, Stadler PF, Schmidt T, Gabaldón T, Lehrach H, Weisshaar B, Himmelbauer H | 2014 | The genome of the recently domesticated crop plant sugar beet (Beta vulgaris) | ftp://ftp.ncbi.nlm.nih.gov/genomes/all/GCA/000/511/025/GCA_000511025.2_RefBeet-1.2.2 | NCBI genome, RefBeet-1.2 |
| Kawahara Y, de la Bastide M, Hamilton JP, Kanamori H, McCombie WR, Ouyang S, Schwartz DC, Tanaka T, Wu J, Zhou S, Childs KL, Davidson RM, Lin H, Quesada-Ocampo L, Vaillancourt B, Sakai H, Lee SS, Kim J, Numa H, Itoh T, Buell CR, Matsumoto T | 2013 | Improvement of the Oryza sativa Nipponbare reference genome using next generation sequence and optical map data | http://rice.plantbiology.msu.edu/pub/data/Eukaryotic_Projects/o_sativa/annotation_dbs/pseudomolecules/version_7.0/all.dir/ | Rice Genome Annotation Project, Rice Gene Models in Release 7 |
| Mayer KF, Waugh R, Brown JW, Schulman A, Langridge P, Platzer M, Fincher GB, Muehlbauer GJ, Sato K, Close TJ, Wise RP, Stein N | 2012 | A physical, genetic and functional sequence assembly of the barley genome. | ftp://ftp.ensemblgenomes.org/pub/plants/release-45/fasta/hordeum_vulgare | Ensembl Genomes, IBSC_v2 |
| Fernandez-Pozo N, Menda N, Edwards | 2015 | The Sol Genomics Network (SGN)– from genotype to phenotype to | ftp://ftp.solgenomics.net/tomato_genome/an- | Sol Genomics Network, ITAG2.4 |

JD, Saha S, Tecle IY, Strickler SR, Bombarely A, Fisher-York T, Pujar A, Foerster H, Yan A, Mueller LA

breeding

notation/ITAG2.4_re-lease/

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
