## [Decision Letter]

**Acceptance summary:**

Plant NLR proteins are immune receptors that activate defenses upon recognition of pathogen proteins. We have recently learned that in the coiled-coil NLR (CC-NLR) protein ZAR1 upon activation the N terminal segment forms a what appears to be a pore that can make holes into the plasma membrane, thereby triggering both cell death and immune responses.

Whether this finding applies to other NLRs remains unknown. The current study shows that a surprisingly large number of plant NLRs share a similar N terminal motif as found in ZAR1. This motif, termed MADA motif, primarily is found in CC-NLRs that execute defense signaling, but seems to be degenerated in CC-NLRs that solely function in sensing pathogens. The authors further show that this motif is functional and interchangeable between ZAR1 and other NLRs. Although some of the details still need to be investigated further, these findings already in the current state significantly advance our understanding of this class of plant immune receptors.

**Decision letter after peer review:**

Thank you for submitting your article "An N-terminal motif in NLR immune receptors is functionally conserved across distantly related plant species" for consideration by *eLife*. Your article has been reviewed by three peer reviewers, including Jian-Min Zhou as the Reviewing Editor and Reviewer #1, and the evaluation has been overseen by Detlef Weigel as the Senior Editor.

The reviewers have discussed the reviews with one another and the Reviewing Editor has drafted this decision to help you prepare a revised submission.

Summary:

Transposon-mutagenesis combined with YFP-fusion experiments suggested that a minimal of 29 amino acids from the N terminus of NRC4, a CC-NLR, can trigger cell death when overexpressed in *N. benthamiana* plants. Extensive bioinformatics analyses identified a conserved motif, dubbed MADA motif, in the very N terminus of ~20% CC-NLRs. Interestingly, the MADA motif is harbored by the N-terminal α-helix of ZAR1, which has been shown to act as a molecular switch in triggering cell death and immunity. Site-directed mutagenesis and motif-swapping experiments nicely supported exchangeability of MADA motif among different CC-NLRs. Together, the authors provide evidence that the N-terminal switch model may generally apply to a large number of CC-NLRs.

Essential revisions:

1) In ZAR1, the MADA motif corresponds to the N-terminal helix that rearranges upon oligomer formation into the part of the funnel that protrudes away from the rest of the pentamer, has a very hydrophobic exterior and was proposed to interact with the membrane. It is not explained how a 29 amino acid peptide could function by itself. It is also difficult to imagine a fusion to GFP could lead to a pentameric configuration. This concern should be addressed prior to publication in *eLife*. Does the 29 amino acid peptide or intact NRC4 protein (activated form) oligomerize? Do they associate with plasma membrane? Any mechanistic data would help greatly.

2) Non-MADA motifs, such as ones from tribes 1, 3, and 4, should be included in the motif-swapping experiment. Maybe MADA motif is not special?

[Editors' note: further revisions were requested prior to acceptance, as described below.]

Thank you for resubmitting your work entitled "An N-terminal motif in NLR immune receptors is functionally conserved across distantly related plant species" for further consideration by *eLife*. Your revised article has been reviewed by three peer reviewers, including Jian-Min Zhou as the Reviewing Editor, and the evaluation has been overseen by Detlef Weigel as the Senior Editor.

The manuscript has been improved but there are some remaining issues that need to be addressed before acceptance, as outlined below:

1) The introduction of a mutation in the YFP protein that prevents its self-association abrogated the observed activities of the MADA motif. While this experiment indirectly supports the hypothesis that the motif functions by self-associating, direct evidence is lacking.

2) The MADA motif peptide fused to YFP forms puncta in the cytoplasm, and some of which probably localize to the plasma membrane. Clear evidence showing membrane binding of the MADA motif fusion protein remains to be provided.

3) The finding that Rx (residues 1-17) confers cell death activity, although it does not contain a MADA motif, challenges the proposed division between sensor NLRs and helper NLRs. Although it is possible that "NLR-sensors may have varying levels of MADA N-termini degeneration", as stated in the manuscript, other possibilities remain.

The reviewers are of the opinion that future research is needed to fully resolve these issues, with the experiments required for this being beyond the reach of the current study. Having said that, the current study provides a nice computational and functional analyses of N terminal domain of CC NLRs, and the findings are of wide interest given the emerging central role of NLR CC domains in plant immunity. We therefore would be happy to accept a revised manuscript in which the authors thoroughly discuss the pitfalls mentioned above and clearly point to future studies that are needed to resolve ambiguous questions about CCs and the MADA motif.

---

## [Author Response]

Essential revisions:1) In ZAR1, the MADA motif corresponds to the N-terminal helix that rearranges upon oligomer formation into the part of the funnel that protrudes away from the rest of the pentamer, has a very hydrophobic exterior and was proposed to interact with the membrane. It is not explained how a 29 amino acid peptide could function by itself. It is also difficult to imagine a fusion to GFP could lead to a pentameric configuration. This concern should be addressed prior to publication in eLife. Does the 29 amino acid peptide or intact NRC4 protein (activated form) oligomerize? Do they associate with plasma membrane? Any mechanistic data would help greatly.

Previous structural analysis of GFP complex revealed that fluorescent proteins form high-order complexes including pentamers (Kim et al., 2015). Thus, we hypothesized that YFP promotes oligomer formation of the NRC4 29 amino acid peptide. To test this hypothesis, we mutated YFP alanine 206 to lysine. This is a common mutation reducing YFP homo-affinity, and is often called “monomeric mutation” (Zacharias et al., 2002). Consistent with our hypothesis, the A206K mutation compromised cell death triggered by NRC4_1-29_-YFP (Figure 2—figure supplement 2).

We have also performed experiments to get clues to the second question ‘Do they associate with plasma membrane?’. We observed subcellular localization of NRC4_1-29_-YFP in *N. benthamiana* leaves and found that the NRC4_1-29_-YFP fluorescence forms puncta in the cytoplasm associated with lower accumulation in the nucleus (Figure 9). Interestingly, some of the puncta were probably at the plasma membrane (Figure 9). We still do not know what are these NRC4_1-29_-YFP puncta, but the MADA and YFP A206K mutations reduced puncta formation (Figure 9). These results indicate that both intact MADA motif and YFP are required for the formation of puncta, which might be associated with the membrane. Consistent with this view, we have also confirmed that MADA mutations suppress NRC4_1-29_-YFP cell death (Figure 7—figure supplement 1). We believe that these results are consistent with the current resistosome model and provide a useful platform for future NLR research.

2) Non-MADA motifs, such as ones from tribes 1, 3, and 4, should be included in the motif-swapping experiment. Maybe MADA motif is not special?

We have addressed this request and we now include additional swapping experiments. We swapped the N-termini of Rx, Bs2 and LOV1 into NRC4 autoactive mutant background. LOV1, Bs2 and Rx N-terminal domains belonged to N-terminal tribes 2, 11, and 25, respectively, and they were not detected as MADA-NLRs in our HMMER analysis. Rx and Bs2 are NRC-dependent sensor NLR, whereas LOV1 is the only non-MADA-NLR in *Arabidopsis* RPP8 gene cluster (mentioned in the Discussion). Bs2_1-17_ and LOV1_1-25_ sequences could not confer cell death activity to NRC4 autoactive mutant (Figure 10), however, Rx_1-17_ did. We have confirmed that the phenotypes are not due to differences in protein accumulation. We included the results and discussed their implications in the Discussion section. We conclude that NLR-sensors may have varying levels of MADA N-termini degeneration.

[Editors' note: further revisions were requested prior to acceptance, as described below.]

The manuscript has been improved but there are some remaining issues that need to be addressed before acceptance, as outlined below:1) The introduction of a mutation in the YFP protein that prevents its self-association abrogated the observed activities of the MADA motif. While this experiment indirectly supports the hypothesis that the motif functions by self-associating, direct evidence is lacking.

We added the following text to the Discussion:

“We also found that mutations that perturb the MADA motif and prevent YFP self-association impair the capacity of NRC4_1-29_-YFP to cause cell death and form puncta in *N. benthamiana* leaf cells (Figure 2—figure supplement 2, Figure 7—figure supplement 1, Figure 9). […] In the future, further biochemical, structural and cellular analyses are needed to determine the precise nature of the broadly conserved MADA motif and address the extent to which the ZAR1 “death switch” model occurs in CC-NLRs.”

2) The MADA motif peptide fused to YFP forms puncta in the cytoplasm, and some of which probably localize to the plasma membrane. Clear evidence showing membrane binding of the MADA motif fusion protein remains to be provided.

The text above also addresses this point.

Please note that we observed dynamics of the NRC4_29_-YFP puncta in living cells under microscope. The puncta might be translocated to not only plasma membrane also the other subcellular membrane compartments. We think that addressing in details the subcellular translocation in living cells requires more detailed analyses and is beyond the scope of the main finding of this manuscript. We stated the unanswered question as a future perspective in the Discussion.

3) The finding that Rx (residues 1-17) confers cell death activity, although it does not contain a MADA motif, challenges the proposed division between sensor NLRs and helper NLRs. Although it is possible that "NLR-sensors may have varying levels of MADA N-termini degeneration", as stated in the manuscript, other possibilities remain.

We have one paragraph in the Discussion devoted to the Rx exception to the correlation between MADA predictions and functional complementation of NRC4. In addition, we included this text to the paragraph:

“In the future, it would be fascinating to determine resistosome configurations of NLR sensor and helper hetero-complexes. As discussed elsewhere (Adachi et al., 2019a; Dangl review), one hypothesis is that sensor NLRs associate with the resistosome as functional equivalents of RLCKs in the ZAR1 resistosome. […] Future structural analyses of NLR sensor/helper heterocomplexes are needed to address these questions.”